# A Physics-Informed Neural Network approach for compartmental epidemiological models

**Caterina Millevoi** [1] *, **Damiano Pasetto** [2], **Massimiliano Ferronato** [1]

**1** Department of Civil, Environmental and Architectural Engineering, University of Padova, via Marzolo 9, Padova, Italy, **2** Department of Environmental Sciences, Informatics and Statistics, Ca' Foscari University of Venice, Via Torino 155, Venezia Mestre, Italy

* caterina.millevoi@unipd.it

## Abstract

Compartmental models provide simple and efficient tools to analyze the relevant transmission processes during an outbreak, to produce short-term forecasts or transmission scenarios, and to assess the impact of vaccination campaigns. However, their calibration is not straightforward, since many factors contribute to the rapid change of the transmission dynamics. For example, there might be changes in the individual awareness, the imposition of non-pharmacological interventions and the emergence of new variants. As a consequence, model parameters such as the transmission rate are doomed to vary in time, making their assessment more challenging. Here, we propose to use Physics-Informed Neural Networks (PINNs) to track the temporal changes in the model parameters and the state variables. PINNs recently gained attention in many engineering applications thanks to their ability to consider both the information from data (typically uncertain) and the governing equations of the system. The ability of PINNs to identify unknown model parameters makes them particularly suitable to solve ill-posed inverse problems, such as those arising in the application of epidemiological models. Here, we develop a reduced-split approach for the implementation of PINNs to estimate the temporal changes in the state variables and transmission rate of an epidemic based on the SIR model equation and infectious data. The main idea is to split the training first on the epidemiological data, and then on the residual of the system equations. The proposed method is applied to five synthetic test cases and two real scenarios reproducing the first months of the Italian COVID-19 pandemic. Our results show that the split implementation of PINNs outperforms the joint approach in terms of accuracy (up to one order of magnitude) and computational times (speed up of 20%). Finally, we illustrate that the proposed PINN-method can also be adopted to produced short-term forecasts of the dynamics of an epidemic.

## Author summary

During the recent COVID-19 pandemic, we all became familiar with the reproduction number, a crucial quantity to determine if the number of infections is going to increase or decrease. Understanding the past changes of this quantity is fundamental to produce

**Data Availability Statement:** The epidemiological data for the Italian COVID-19 epidemic are available at the following link: https://www.epicentro.iss.it/en/coronavirus/sars-cov-2-integrated-surveillance-data. The source code in Python is available at the

following repository: https://github.com/cmillevoi/EpiPINN.

**Funding:** The author(s) received no specific funding for this work.

**Competing interests:** The authors have declared that no competing interests exist.

realistic forecasts of the epidemic and to plan possible containment strategies. There are several methods to infer the values of the reproduction number and, thus, the number of new infections. Statistical methods are based on the analysis of the collected epidemiological data. Instead, modeling approaches (such as the popular SIR model) attempt constructing a set of mathematical equations whose solution aims at approximating the dynamics underlying the data.

In this paper, we explore the use of a recently developed technique called Physics-Informed Neural Network, which tries to combine the two approaches and to simultaneously fit the data, infer the dynamics of the unknown parameters, and solve the model equations.

The proposed PINN implementations are tested in different scenarios using both synthetic and real-world data referred to the COVID-19 pandemic outbreak in Italy. The promising results can pave the way for a wider use of PINNs in epidemiological applications.

## 1 Introduction

Epidemiological models are nowadays fundamental to assist and guide policy makers in the fight against the spreading of diseases. This has been evident during the recent COVID-19 pandemic, when epidemiologists and scientists all over the world devoted their research to develop ad-hoc transmission models. Focusing, for example, on Italy, where the European outbreak started in February 2020, epidemiological models have been adopted to analyze different aspects of the epidemic: to determine the urgency to impose regional restrictions [1]; to analyze the impact of the national lockdown [2, 3]; to explore the results of transmission scenarios after the release of the restrictions [4]; to study the impact of the different variants and the vaccination campaign [5–7]; and to compute optimal strategies for the vaccine deployment in order to minimize the number of cases or deaths [8, 9]. Most of these studies describe the SARS-CoV-2 transmission using different variations of compartmental models. The basic SIR model is at the core of those more-complex epidemiological models. It subdivides the population of interest into compartments indicating the infectious status of each individual (i.e. susceptible, infected, or recovered individuals). The dynamic describes the mean contacts between susceptible and infected individuals, and thus, the average rate at which susceptible individuals transit to the infected compartment. The main model parameter is the rate of transmission of the infection, $\beta$. This is strictly related to the well known basic reproduction number, $\mathcal{R}_0$, representing the average number of secondary infections generated by one infected individual in a totally susceptible population. The value of this quantity changes during an outbreak due to the temporal variations in human behavior (caused, for example, by changes in individual awareness or social distancing policies) and in the infectiousness of the virus. The effective reproduction number, $\mathcal{R}_t$, aims at describing the ongoing transmission in a changing system.

Data-driven methods provide effective estimates of $\mathcal{R}_t$ based on the renewal equation [10–12], i.e., a convolution on the reported cases having as kernel the serial interval (the time interval between the symptom onset of an individual and its secondary infections). These data-driven estimates do not explicitly provide a relationship between the changes in $\mathcal{R}_t$ and its possible causes, such as the implemented non-pharmaceutical interventions or the vaccination campaigns. Compartmental models give a deeper understanding of the ongoing spreading of the disease and, at the same time, allow the computation of $\mathcal{R}_t$ using the spectral radius of the

next generation matrix [13–15]. However, they require the assessment and calibration of time-dependent parameters.

Tracking the temporal variations in the model parameters is an essential but complex problem to follow and predict the spreading of a disease. Many studies tackle this problem using Bayesian inference, i.e., searching for the posterior distribution of the unknown parameters based on the available reported cases and the prior distribution. Among these approaches, we recall the iterative particle filter [16], sequential data-assimilation schemes [17], or the use of subsequent Markov chain Monte Carlo (MCMC) [4, 7]. Being based on random sampling, these approaches might result in low quality results and large computational times, due to the slow Monte Carlo convergence.

Here, we propose to adopt a deterministic approach based on Physics-Informed Neural Networks (PINNs). The idea behind PINNs is to exploit the universal approximation property of Neural Networks (NNs) [18, 19] to estimate the solution of differential equation [20]. In practice, this is done by describing the state variables and, in case, the time-dependent parameters using NNs. The parameters of the NNs are trained by seeking the minimum of a loss function based on both the misfit on the available data, and the residual of the differential equations governing the problem at hand, i.e., the SIR model equations in our case. Thus, the PINN functions fit the data and, at the same time, provide good approximations of the solutions of the differential equations. The use of the epidemiological model equations is fundamental in PINNs and constitutes the main innovation with respect to simpler NNs or Deep Neural Networks (DNNs), which are completely data-driven.

The application of PINNs to epidemiological models became particularly relevant during the COVID-19 pandemic. Many studies used PINNs as an inverse-problem solver, to calibrate the parameters of epidemiological compartmental models. However, the model parameters has frequently been considered constant in time, e.g, [21, 22], or with particular periodic dependencies on time [23]. Schiassi et al. [24] showed the computational efficacy of using PINNs to estimate constant parameters of different basic compartmental models under increasing levels of noise in the data. Long et al. [25] considered a more realistic scenario, and used PINNs to accurately identify the time-varying transmission parameter in a SIRD model of COVID-19 when assimilating the reported infected cases in three USA states. Feng et al. [26] proposed a similar approach to predict the number of active cases and removed cases in the US. Olumoyin et al. [27] used PINNs to track the changes in transmission rate and the number of asymptomatic individuals for COVID-19. Ning et al. [28] and He et al. [29] presented applications of PINNs to COVID-19 outbreaks in Italy and China, respectively. Bertaglia et al. [30] constrained PINNs to satisfy an asymptotic-preservation property to avoid poor results caused by the multiscale nature of the residual terms in the loss function.

Building on top of these examples, our work aims to deeper explore the properties of PINNs as an inverse solver for the estimation of time-dependent transmission rates or reproduction numbers in SIR models. Our analysis aims on further showing some benefits of using PINNs that are not directly available with more traditional approaches such as: the simultaneous estimation of multiple parameters that change in time, the inference using jointly different types of data, the possibility of providing a future projection for the evaluated parameters, the possibility of training the model even if there are gaps or large errors or uncertainties on the quality of the data.

In particular, we propose two modifications of the PINNs algorithm that grant faster convergence and more stable results, thus providing a step forward in the use of PINNs in real epidemiological models.

The first modification splits the PINN implementation in two steps. The motivation for this approach is that in the common PINN implementation for SIR-like models, the NNs

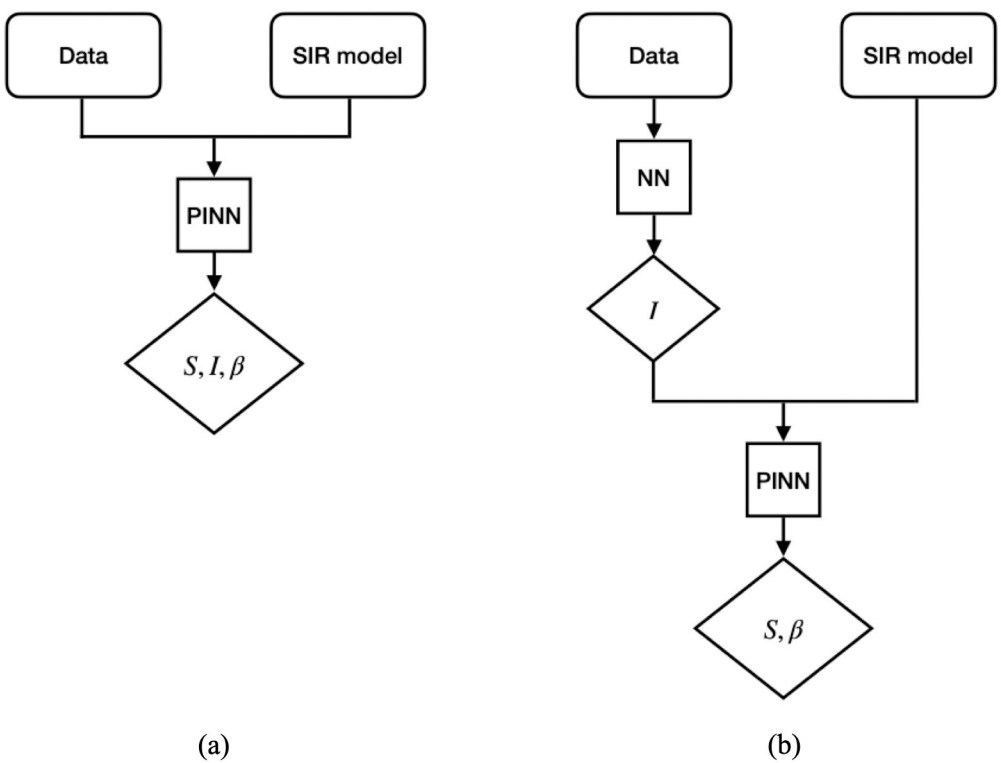

**Fig 1. Diagram of the workflow of the joint (a) and split (b) PINN approaches.**

representing the model state variables and, if present, the time-dependent parameters, are calibrated together through the minimization of the loss function on the data and the model residual. This inverse problem is particularly complex and many epochs might be required to achieve convergence. Starting from the idea that the available epidemiological data, which are typically the daily or weekly reported infections, is directly associated to a model state variable, the split PINN approach is based on the following two steps: as first, construct the NN of the state variable associated to the data, e.g., the infected compartment, by minimizing the loss function based on the data; as second, calibrate the other NNs for the remaining state variables and parameters based on the NN computed in the first step and the minimization of the residuals of the governing equations. We will refer to the traditional PINN approach as *joint* approach, in contrast to the described *split* approach. A graphical sketch of the two approaches is shown in Fig 1.

The second proposed modification reduces the number of NNs considered in the PINN approximation and, consequently, simplifies the structure of the loss function. This simplification is possible because, in simple SIR-based models, the transmission parameter and the infected compartment control the system dynamic. In fact, these functions allow to directly evaluate the other state variables, which are then redundant in the formulation of the loss function.

Our analysis compares the joint, split, and reduced approaches in a sequence of synthetic test cases where we progressively challenge the structure of the transmission rate from constant, to a sinusoidal-like dependence on time, to a real scenario, and increase the noise on the synthetic reported data. The proposed test cases assume model parameters that are inspired by the first months of the COVID-19 outbreak in Italy. As an example of application, the PINN

strategies are adapted in order to fit the real epidemiological data reported in Italy. Due to the large uncertainties that characterize the real data on the reported infections, in this last setting we propose to include in the loss function also the data on the daily hospitalizations, which are a more reliable representation of the number of individuals with severe symptoms. Finally, we consider this scenario to explore the accuracy of the short-term forecasts produced by PINNs.

The paper is organized as follows. Section 2 presents the mathematical formulation of the proposed methods. It starts with the equations of the SIR model (Section 2.1), then it describes the joint and split implementations of PINNs (Section 2.2), and finishes with the modified schemes for the reduced approach (Section 2.3) and the extension to the hospitalized data (Section 2.4). The numerical results of the application of the proposed PINNs to seven test cases are illustrated in Section 3. The method is tested and validated on synthetic cases (Section 3.1) and then applied in a real-world scenario (Section 3.2) for both parameter estimation and forecast. Finally, Section 4 presents the discussion of the results and sums up the main conclusions.

## 2 Methods

### 2.1 The basic SIR model

The well-known SIR model is largely adopted for the theoretical analysis of epidemics, and lies at the core of several more complex epidemiological models for real applications. At a given time $t$ [T], the individuals in a population of dimension $N$ [–] are subdivided into compartments on the basis of their epidemiological status, in this case the susceptible ($S$), the infected ($I$), and the recovered ($R$) individuals. The number of individuals in the three compartments changes in time under the assumption that, in a well mixed population, any susceptible individual can enter in contact with any infected individual, thus possibly becoming infected itself.

From a mathematical point of view, the strong form of the ordinary differential problem governing these dynamics can be stated as follows. Let $\mathcal{T} = [t_0, t_f] \subseteq \mathbb{R}^+ \cup \{0\}$ be the time domain of interest, with $t_0$ and $t_f$ [T] the initial and final times of the simulation, respectively. Given the continuous functions $\beta(t) : \mathcal{T} \to \mathbb{R}^+$ and $\delta(t) : \mathcal{T} \to \mathbb{R}^+$, find $S(t) : \mathcal{T} \to [0, N]$, $I(t) : \mathcal{T} \to [0, N]$, and $R(t) : \mathcal{T} \to [0, N]$ such that:

$$\begin{cases} \dot{S}(t) = -\dfrac{\beta}{N} I(t) S(t) \\[2mm] \dot{I}(t) = \dfrac{\beta}{N} I(t) S(t) - \delta I(t) \\[2mm] \dot{R}(t) = \delta I(t) \end{cases} \quad , \quad \forall\, t \in \mathcal{T}, \tag{1}$$

and satisfying the initial conditions:

$$\begin{cases} S(t_0) = N - I_0 \\[1mm] I(t_0) = I_0 \\[1mm] R(t_0) = 0 \end{cases} \tag{2}$$

In Eqs (1) and (2) $\beta$ [$\mathrm{T}^{-1}$] is the transmission rate controlling the average rate of the infection, $\delta$ [$\mathrm{T}^{-1}$] is the mean rate of removal of the infected individuals that become recovered. Another relevant quantity used to set up the model is $D = \delta^{-1}$, i.e., the mean reproduction period [T] representing the average time spent by an individual in compartment $I$. Initial conditions for the spreading of a new disease assume that the population at the initial time is

completely susceptible besides a small number $I_0$ of infected individuals (typically 1, but not necessarily).

The basic reproduction number $\mathcal{R}_0$ [-] associated to this model reads $\mathcal{R}_0 = \beta(t_0)/\delta(t_0)$ and provides an estimate of the number of secondary infections generated by one infectious individual in a susceptible population, i.e. at the beginning of the epidemic. The threshold $\mathcal{R}_0 > 1$ indicates the occurrence of an outbreak, while $\mathcal{R}_0 < 1$ indicates that the number of infected individuals is rapidly decreasing. Note that, in a real population, the number of individuals in each compartment is a discrete variable, whose dynamic can be described by stochastic approaches, e.g., the Gillespie method or discrete Markov chains. Hence, the continuous deterministic model in Eqs (1) and (2) is a valuable representation of the mean process in large populations.

Standard numerical ODE solvers, such as Runge-Kutta-based methods, can provide an accurate solution to the differential problem (1) and (2). For $\mathcal{R}_0 > 1$ and constant parameters, the solution depicts an initial exponential-like increase in the number of infections up to a peak, and then a fast decrease due to the depletion of susceptible individuals. However, it is clear that this dynamic does not correspond to what happens during an outbreak. The main challenge when using a model based on (1) to describe a real epidemic is that the transmission rate $\beta$ and the mean reproduction period $\delta^{-1}$ can change in time because of many factors: social behaviors (individual awareness, increase or decrease of gatherings, mobility, social distancing), non-pharmaceutical interventions (use of devices that reduce transmission—such as masks, introduction of lock-downs), changes in the pathogen infectiousness due to new variants, reduction of the susceptibility of the population due to vaccination campaigns. In this evolving scenario, the effective reproduction number $\mathcal{R}_t$ [-] is the critical quantity that controls the spreading of the disease. $\mathcal{R}_t$ is the equivalent of $\mathcal{R}_0$ in time, i.e., $\mathcal{R}_t = \beta(t)/\delta(t) \cdot S(t)/N$, taking into account that the number of susceptible individuals decreases and the main parameters controlling the spreading of the disease generally change. An essential element for a reliable simulation is therefore the assessment of $\mathcal{R}_t$, hence $\beta(t)$ and $\delta(t)$ along with the compartment $S(t)$, from the available epidemiological data. In the following we will assume that $\delta$ is constant in time, assumption done in many epidemiological applications (see e.g., [2, 4, 5]).

## 2.2 PINN solution to the SIR model

Here we develop and analyze a PINN-based approach to simultaneously solve the problem (1) and (2) and estimate the temporal values of the reproduction number by using a time series of infectious individuals as basic epidemiological information.

A standard NN aims to reconstruct an unknown function $u$ from the knowledge of some training data points. The NN approximating a generic $u$, denoted throughout this work by $\hat{u}$, is the recursive composition of the function:

$$\Sigma^{(l)}(\mathbf{x}^{(l)}) = \phi^{(l)}.(\mathbf{W}^{(l)}\mathbf{x}^{(l)} + \mathbf{b}^{(l)}), \tag{3}$$

where $\mathbf{W}^{(l)} \in \mathbb{R}^{n_l \times n_{l-1}}$, $\mathbf{b}^{(l)} \in \mathbb{R}^{n_l}$, and $\phi^{(l)}$ are weights, biases, and activation functions of the $l$-th layer, respectively. The last layer is the output layer, the others are the hidden layers. We denote with $n_l$ the number of neurons in layer $l$. Activation functions are user-specified functions with limited range, which are generally non linear in order to provide a source of non linearity to the NN and maintain low weight values. The Matlab-inspired notation $\phi.(\mathbf{x})$ means that the function $\phi$ is applied to each component of the vector $\mathbf{x}$. Let $L$ be the number of hidden layers. If $u : \mathcal{T} \to \mathbb{R}$ is the solution of an ordinary differential equation in the domain $\mathcal{T}$, the input of the first layer reads $\mathbf{x}^{(1)} = t \in \mathcal{T}$, so $n_0 = 1$, and the output of the last layer $\hat{u}$ is a scalar,

so $n_{L+1} = 1$. Then, the NN for $u$ formally reads:

$$\hat{u}(t) = \Sigma^{(L+1)} \circ \Sigma^{(L)} \circ \cdots \circ \Sigma^{(1)}(t). \tag{4}$$

The NN depends on the set of weights and biases, which are trained through an optimization algorithm so as to minimize an appropriate loss function defined as the mean squared error of $\hat{u}$ over the set of training points. In the case of PINNs, the information from the governing equations of the physical system is introduced in a weak way in the loss function by adding the residual of the differential equations evaluated at some collocation points [31, 32].

For the SIR model (1) and (2), we assume that the training data points for the fitting are the reported infections. Let $\tilde{I}_j$ be the number of reported infected individuals at times $\tilde{t}_j, j = 1, \ldots, N_D$. This might be subject to reporting errors, thus, in general $\tilde{I}_j \neq I(\tilde{t}_j)$. The residual of the governing equations is computed over $N_C$ collocation points.

We aim at finding a NN representation for the susceptible, infected, and recovered individuals ($\hat{S}(t)$, $\hat{I}(t)$, and $\hat{R}(t)$, respectively) along with the transmission rate ($\hat{\beta}(t)$). Since the state variables $S$, $I$, $R$ span an extremely wide range of values (from zero to the population size $N > 10^6$), the functional search is optimized by a proper scaling:

$$S(t) = CS_s(t_s), \quad I(t) = CI_s(t_s), \quad R(t) = CR_s(t_s), \tag{5}$$

where $C$ [-] is an appropriate constant and $t_s$ is the dimensionless scaled temporal variable, $t_s = (t - t_0)/(t_f - t_0)$. The system of ODEs (1) for the scaled variables becomes:

$$\begin{cases} \dot{S}_s(t_s) = -C_1 \beta_s(t_s) I_s(t_s) S_s(t_s) \\ \dot{I}_s(t_s) = C_1 \beta_s(t_s) I_s(t_s) S_s(t_s) - C_2 I_s(t_s) \quad , \quad t_s \in [0, 1], \\ \dot{R}_s(t_s) = C_2 I_s(t_s) \end{cases} \tag{6}$$

where $\beta_s(t_s) : [0, 1] \to \mathbb{R}^+$, $C_1 = (t_f - t_0)C/N$ and $C_2 = (t_f - t_0)\delta C$. The initial conditions (2) are correspondingly scaled as well as the infectious data $\tilde{I}_j = C\tilde{I}_{s,j}$ at times $\tilde{t}_{s,j} = (\tilde{t}_j - t_0)/(t_f - t_0)$.

The SIR model (6) does not consider death and birth processes and assumes a negligible mortality rate of the disease. Thus, the total population $N$ is constant in time and equal to $N = S + I + R$. Under these hypotheses, the PINN model needs only two NNs representing the behavior of the population: one for the state variable of the susceptible individuals $\hat{S}_s$, and one for the infected individuals $\hat{I}_s$. The number of recovered individuals is computed as $\hat{R}_s = \frac{N}{C} - \hat{I}_s - \hat{S}_s$. A third NN is included for the estimation of the transmission rate $\hat{\beta}_s$. In this way, the number of parameters to be tuned during the training is consistently reduced.

It is important to underline that the state variables represent the number of individuals in a compartment, thus they all have positive outputs. Training the model without imposing this condition could lead to nonphysical negative NN outputs. The non-negative constraint can be imposed in the NN in two alternative ways: inserting a penalty term for the negative values of the NNs (weak constraint) or building the NN architecture so as to allow for positive values only (hard constraint). The latter prescription can be met by setting the output activation function, i.e., the one related the last layer, $\phi^{(L+1)}$, equal for example to the square function. An experimental comparison between the two approaches shows that the latter is generally more effective and provides more robust results. The numerical outcomes that follow are therefore obtained by using the hard constraint prescription for the non-negativity of the solution. The same constraint is adopted to entail a positive value for $\beta_s$.

The selection of the loss function is one of the most sensitive steps in the PINN approach, given the multi-objective nature of the method. Using the Mean Squared Error (MSE) as loss measure, the objective is to minimize the mismatch on the $N_D$ data:

$$\mathcal{L}_D(\hat{I}_s) = \omega_D \frac{1}{N_D} \sum_{j=1}^{N_D} [\hat{I}_s(\tilde{t}_{s,j}) - \tilde{I}_{s,j}]^2, \tag{7}$$

the squared norm of the residual of Eq (6) evaluated on $N_C$ collocation points $\{\bar{t}_{s,i}\}_{i=1}^{N_C}$:

$$\mathcal{L}_{ODE}(\hat{S}_s, \hat{I}_s, \hat{\beta}_s) = \frac{1}{N_C} \sum_{i=1}^{N_C} \omega_S \left[ \frac{d\hat{S}_s}{dt_s} + C_1 \hat{\beta}_s \hat{I}_s \hat{S}_s \right]^2 \Big|_{\bar{t}_{s,i}} +$$
$$\frac{1}{N_C} \sum_{i=1}^{N_C} \omega_I \left[ \frac{d\hat{I}_s}{dt_s} - C_1 \hat{\beta}_s \hat{I}_s \hat{S}_s + C_2 \hat{I}_s \right]^2 \Big|_{\bar{t}_{s,i}} + \tag{8}$$
$$\frac{1}{N_C} \sum_{i=1}^{N_C} \omega_R \left[ \frac{d\hat{R}_s}{dt_s} - C_2 \hat{I}_s \right]^2 \Big|_{\bar{t}_{s,i}},$$

and the misfit on the initial conditions:

$$\mathcal{L}_{IC}(\hat{S}_s, \hat{I}_s) = \omega_{S_0} \left[ \hat{S}_s(0) - \frac{N-I_0}{C} \right]^2 + \omega_{I_0} \left[ \hat{I}_s(0) - \frac{I_0}{C} \right]^2 + \omega_{R_0} \hat{R}_s^2(0), \tag{9}$$

where $\omega_*$ are proper weights needed to balance the relative importance of the entries arising from each contribution to the global MSE value. Fig 2 shows a diagram of the PINN implementation for the solution of the scaled SIR model (6).

We explore two possible approaches for the construction of the PINN model, indicated as *joint* or *split*. The joint approach aims to simultaneously calibrate $\hat{S}_s$, $\hat{I}_s$, and $\hat{\beta}_s$ by minimizing

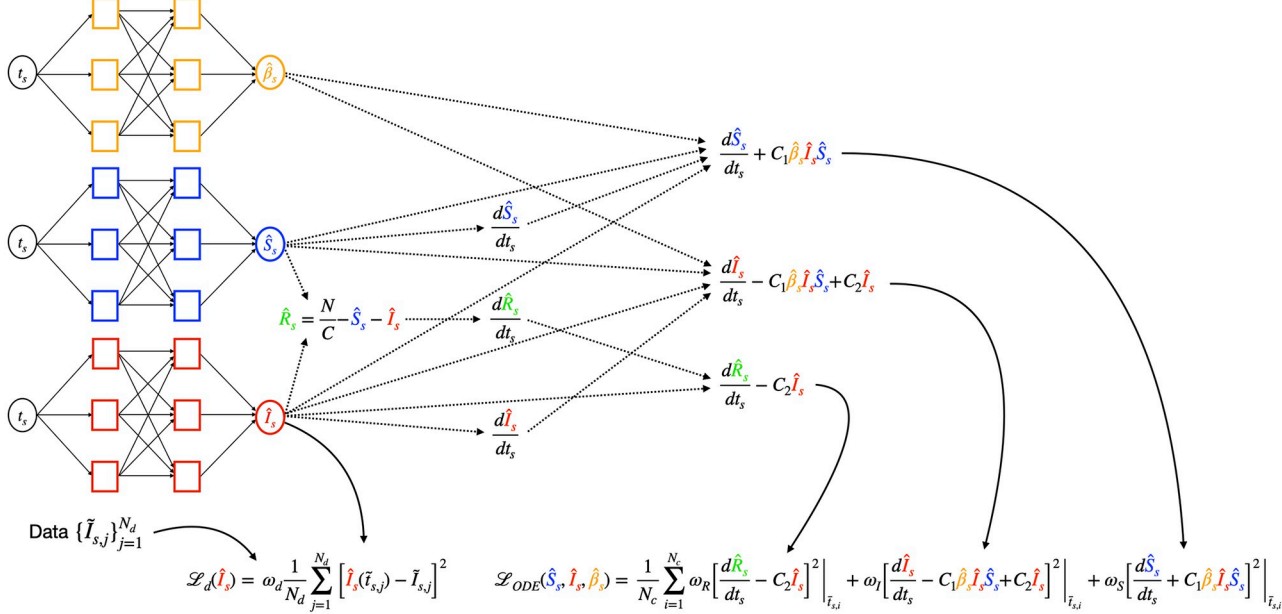

**Fig 2. Diagram of the PINN model for the SIR equations with unknown $\beta(t)$.** The parameters of the NNs for $\beta$, $S$, $I$ are obtained by minimizing the loss functions on the infectious data, and on the residual and initial conditions of the model equations.

the joint loss function corresponding to the sum of $\mathcal{L}_D$, $\mathcal{L}_{ODE}$, and $\mathcal{L}_{IC}$:

$$\mathcal{L}_{\text{joint}}(\hat{S}_s, \hat{I}_s, \hat{\beta}_s) = \mathcal{L}_D(\hat{I}_s) + \mathcal{L}_{ODE}(\hat{S}_s, \hat{I}_s, \hat{\beta}_s) + \mathcal{L}_{IC}(\hat{S}_s, \hat{I}_s). \tag{10}$$

By distinction, the split approach subdivides the overall problem. First, $\hat{I}_s$ is independently calibrated on the data error $\mathcal{L}_D$ (7) only. In this case, a standard NN is used with weight $\omega_D = 1$, thus obtaining a differentiable regression function for the data. The only-data regression is followed by a fully-physics-informed regression, where the parameters defining $\hat{S}_s$ and $\hat{\beta}_s$ are trained by minimizing:

$$\mathcal{L}_{\text{split}}(\hat{S}_s, \hat{\beta}_s) = \mathcal{L}_{ODE}(\hat{S}_s, \hat{\beta}_s) + \mathcal{L}_{IC}(\hat{S}_s). \tag{11}$$

It is important to underline that in standard data-driven NNs a regularization term is frequently added to the loss function to avoid overfitting on the data. The term related to the residual in the loss function (Eq 9 in our case) acts as a regularization in PINNs, therefore no additional regularization has been added (see [20] for more details).

## 2.3 Reduced SIR model

The system of ODEs in (1) can be further reduced by directly considering the definition of the effective reproduction number $\mathcal{R}_t$. By easy developments, the model (1) becomes:

$$\begin{cases} \dot{I}(t) = \delta(\mathcal{R}_t - 1)I(t) \\ \dot{S}(t) = -\delta\mathcal{R}_t I(t) \end{cases}, \quad t \in [t_0, t_f]. \tag{12}$$

where the unknown functions are $I(t)$ and $S(t)$, and the state variable $R(t)$ is simply obtained from the consistency relationship $R(t) = N - S(t) - I(t)$. The initial conditions (2) still hold. The new system (12) can be solved sequentially by integrating the upper equation first and then computing $S(t)$ from the second equation.

This approach reduces the number of functions that are approximated by NNs to two, i.e., $I$ and $\mathcal{R}_t$, and eliminates any redundant term in the loss function minimized in the PINN approach. The same scaling as in Eq (5) is used for the state variable $I$, so that the upper equation in (12) reads:

$$\dot{I}_s(t_s) = \delta(t_f - t_0)(\mathcal{R}_t - 1)I_s(t_s), \quad t_s \in [0, 1]. \tag{13}$$

The NNs approximating the variables of interest, i.e., $\hat{I}_s$ and $\hat{\mathcal{R}}_t$, can be obtained by minimizing the mismatch on data $\mathcal{L}_D$ (Eq (7)) and the squared norm of the residual of Eq (13) on $N_C$ collocation points:

$$\mathcal{L}_{r,ODE}(\hat{I}_s, \hat{\mathcal{R}}_t) = \frac{1}{N_C} \sum_{i=1}^{N_C} \left[ \frac{d\hat{I}_s}{dt_s} - \delta(t_f - t_0)(\hat{\mathcal{R}}_t - 1)\hat{I}_s \right]^2 \bigg|_{\bar{t}_{s,i}} \tag{14}$$

Notice that in this case the contributions in $\mathcal{L}_D$ and $\mathcal{L}_{r,ODE}$ have a consistent size, hence there is no need for introducing the weight parameters $\omega_*$ to balance the loss function terms. For this reason, we simply set $\omega_D = 1$ in the expression (7).

The joint and split approaches can be formulated for this PINN-based model as well. The joint approach consists in training simultaneously the NNs $\hat{I}_s$ and $\hat{\mathcal{R}}_t$ by minimizing the total

loss function:

$$\mathcal{L}_{r,\text{joint}}(\hat{I}_s, \hat{\mathcal{R}}_t) = \mathcal{L}_D(\hat{I}_s) + \mathcal{L}_{r,ODE}(\hat{I}_s, \hat{\mathcal{R}}_t). \tag{15}$$

By distinction, the split approach implies training $\hat{I}_s$ on the data only by the minimization of $\mathcal{L}_D$ in Eq (7). Then, the time-dependent parameter $\hat{\mathcal{R}}_t$ is obtained by minimizing:

$$\mathcal{L}_{r,\text{split}}(\hat{\mathcal{R}}_t) = \mathcal{L}_{r,ODE}(\hat{\mathcal{R}}_t). \tag{16}$$

Notice that in the reduced PINN model no initial condition is set, but we let the model deduce it from the data. From a theoretical viewpoint, initial conditions are not necessary because $\hat{I}_s$ is obtained from the data, while the governing differential Eq (13) is used to calibrate $\hat{\mathcal{R}}_t$. This outcome is relevant because it replicates what typically happens in a real-case scenario, where there is no actual knowledge about the instant of beginning of the outbreak. In fact, the case 0 in most outbreaks is unknown and the conventional start of the epidemic has a number of infected individuals that is usually largely underestimated. The use of the reduced modeling approach makes it possible to remove the term related to the initial condition from the loss function.

## 2.4 SIR model with the hospitalization compartment

The reported infections can be often affected by large uncertainties. Especially at the beginning of an epidemic outbreak, the disease cannot be easily recognized, either because of the difficulty of correctly identifying the symptoms, or the absence of well-established detection and surveillance procedures, or the impossibility of reaching and testing all the people infected by the disease. Moreover, these data can be strongly affected by territorial peculiarities and the logistic of testing facilities. Hence, founding an epidemiological model on these pieces of information can undermine its reliability. A much less uncertain epidemiological datum is the daily number of individuals that require to be hospitalized. This fraction of the overall number of infected individuals is representative of the entire $I$ compartment by assuming that hospitalization is needed over a certain common threshold level of symptoms in the population.

We introduce a new variable, $H$, defined as:

$$H(t) = \delta\sigma I(t), \tag{17}$$

where $\sigma$ represents the fraction of infected individuals moving to the hospitalized compartment. Note that also parameter $\sigma$ might change in time, for example because of the insurgence of more aggressive variants or the improvement of home treatment. A more convenient formulation uses the cumulative number $\Sigma_H$ of hospitalized individuals:

$$\Sigma_H(t) = \int_{t_0}^t H(z)\,\mathrm{d}z. \tag{18}$$

The new formulation of the updated SIR model can be therefore stated as follows. Given $\mathcal{R}_t : \mathcal{T} \to \mathbb{R}^+$, $\sigma(t) : \mathcal{T} \to [0,1]$, and $\delta(t) : \mathcal{T} \to \mathbb{R}^+$, find $\Sigma_H(t) : \mathcal{T} \to [0,N]$, $I(t) : \mathcal{T} \to [0,N]$, and $S(t) : \mathcal{T} \to [0,N]$ such that:

$$\begin{cases} \dot{\Sigma}_H(t) = \delta\sigma I(t) \\ \dot{I}(t) = \delta(\mathcal{R}_t - 1)I(t) \\ \dot{S}(t) = -\mathcal{R}_t\delta I(t) \end{cases} , \quad \forall\, t \in \mathcal{T}, \tag{19}$$

with $R(t) = N - I(t) - S(t)$, the initial conditions (2) and $\Sigma_H(t_0) = 0$. The available information from the actual epidemiological data is the daily variation $\Delta_H$ of the cumulative number of hospitalized individuals:

$$\Delta_H(t) = \Sigma_H(t) - \Sigma_H(t-1) \simeq \dot{\Sigma}_H(t), \tag{20}$$

whose values represents the training dataset for the PINN approximation of system (19). As previously done, the functional search of the approximating NNs is carried out on the properly scaled quantities $I(t) = CI_s(t_s)$ (see Eq (5)) and:

$$\Delta_H(t) = C_H \Delta_{H,s}(t_s), \tag{21}$$

with $C_H$ the scaling factor. The upper equation in system (19) with the scaled quantities reads:

$$C_H \Delta_{H,s}(t_s) = \delta C \sigma_s(t_s) I_s(t_s), \tag{22}$$

with $\sigma_s : \mathcal{T} \to [0, 1]$, while the second scaled equation is the same as in (13). Hence, the NNs needed to solve the SIR model with hospitalization data are $\hat{\Delta}_{H,s}, \hat{I}_s, \hat{\sigma}_s$, and $\hat{\mathcal{R}}_t$. The training data points for the fitting are both the scaled reported infections $\tilde{I}_{s,j}$ and the hospitalizations $\tilde{\Delta}_{H,s,j}$ at the scaled times $\tilde{t}_{s,j}, j = 1, \ldots, N_D$. The NNs can be obtained by minimizing the mismatch (7) on the infection data and on the hospitalization data:

$$\mathcal{L}_H = \frac{1}{N_D} \sum_{j=1}^{N_D} [\hat{\Delta}_{H,s}(\tilde{t}_{s,j}) - \tilde{\Delta}_{H,s,j}]^2, \tag{23}$$

and the squared norm of the residuals of Eqs (13) and (22) on $N_C$ collocation points:

$$\mathcal{L}_{H,ODE}(\hat{\Delta}_{H,s}, \hat{I}_s, \hat{\sigma}_s, \hat{\mathcal{R}}_t) = \frac{1}{N_C} \sum_{i=1}^{N_C} \left[ \frac{d\hat{I}_s}{dt_s} - \delta(t_f - t_0)(\hat{\mathcal{R}}_t - 1)\hat{I}_s \right]^2 \Bigg|_{\tilde{t}_{s,i}} + \\ \frac{1}{N_C} \sum_{i=1}^{N_C} \left[ \hat{\Delta}_{H,s} - \frac{\delta C \hat{\sigma}_s}{C_H} \hat{I}_s \right]^2 \Bigg|_{\tilde{t}_{s,i}}. \tag{24}$$

The joint approach consists in the simultaneous estimate of $\hat{\Delta}_{H,s}, \hat{I}_s, \hat{\sigma}_s$, and $\hat{\mathcal{R}}_t$ by finding the minimum to the functional:

$$\mathcal{L}_{H,joint}(\hat{\Delta}_{H,s}, \hat{I}_s, \hat{\sigma}_s, \hat{\mathcal{R}}_t) = \mathcal{L}_D(\hat{I}_s) + \mathcal{L}_H(\hat{\Delta}_{H,s}) + \mathcal{L}_{H,ODE}(\hat{\Delta}_{H,s}, \hat{I}_s, \hat{\sigma}_s, \hat{\mathcal{R}}_t). \tag{25}$$

As for the PINN solution to the reduced SIR model, it is not necessary to include the mismatch on the initial conditions into the global loss function (25) because they are met through the available training data. Moreover, also the use of non-unitary weights $\omega_*$ for the different contributions to $\mathcal{L}_{H,joint}$ is not required since all terms are likely to have a similar magnitude.

In the split approach, $\hat{\Delta}_{H,s}$ is directly trained with the hospitalization data only by minimizing $\mathcal{L}_H$ in Eq (23). Then, $\hat{I}_s$ is computed from (22) as:

$$\hat{I}_s = \frac{C_H}{\delta C} \frac{\hat{\Delta}_{H,s}}{\hat{\sigma}_s}, \tag{26}$$

and $\hat{\sigma}_s$ and $\hat{\mathcal{R}}_t$ are trained by minimizing:

$$
\begin{aligned}
\mathcal{L}_{H,\text{split}}(\hat{\sigma}_s, \hat{\mathcal{R}}_t) \quad = \quad & \frac{1}{N_D}\sum_{j=1}^{N_D}\left[\frac{C_H}{\delta C}\frac{\hat{\Delta}_{H,s}(\tilde{t}_{s,j})}{\hat{\sigma}_s(\tilde{t}_{s,j})} - \tilde{I}_{s,j}\right]^2 + \\
& \frac{1}{N_C}\sum_{i=1}^{N_C}\left\{\frac{C_H}{C}\left[\frac{d}{dt_s}\left(\frac{\hat{\Delta}_{H,s}}{\delta\hat{\sigma}_s}\right) - (t_f - t_0)(\hat{\mathcal{R}}_t - 1)\frac{\hat{\Delta}_{H,s}}{\hat{\sigma}_s}\right]\right\}^2\Bigg|_{\bar{t}_{s,i}}.
\end{aligned}
\tag{27}
$$

In real-world scenarios, the new daily infections is a more common piece of information than the total number of infected individuals. In order to include these data in the PINN model, we introduce the cumulative number $\Sigma_I$ of infected individuals:

$$
\Sigma_I(t) = \int_{t_0}^{t} I(z)\ \mathrm{d}z.
\tag{28}
$$

The variation of $\Sigma_I$ in time coincides with negative variation of the class of susceptible individuals $S(t)$, so we can simply update the SIR model with hospitalization data (19) by replacing the last equation with:

$$
\dot{\Sigma}_I(t) = \delta\mathcal{R}_t I(t)\ .
\tag{29}
$$

Since the available information is the daily variation $\Delta_I$ of the cumulative number of infected individuals:

$$
\Delta_I(t) = \Sigma_I(t) - \Sigma_I(t-1) \simeq \dot{\Sigma}_I(t),
\tag{30}
$$

we use these values as training data set. As usual, scaled values are considered such as $\Delta_I = C\Delta_{I,s}$ and we assume that the set of scaled values $\tilde{\Delta}_{I,s,j}$ is available at the training scaled times $\tilde{t}_{s,j}, j = 1, \ldots, N_D$, instead of $\tilde{I}_{s,j}$. The mismatch of $\hat{\Delta}_{I,s}$, i.e., the NN approximating $\Delta_{I,s}$, with the data is measured by:

$$
\mathcal{L}_I = \frac{1}{N_D}\sum_{j=1}^{N_D}\left[\hat{\Delta}_{I,s}(\tilde{t}_{s,j}) - \tilde{\Delta}_{I,s,j}\right]^2,
\tag{31}
$$

while the squared norm of the residual reads:

$$
\begin{aligned}
\mathcal{L}_{HI,ODE}(\hat{\Delta}_{H,s}, \hat{\Delta}_{I,s}, \hat{I}_s, \hat{\sigma}_s, \hat{\mathcal{R}}_t) \quad = \quad & \frac{1}{N_C}\sum_{i=1}^{N_C}\left[\frac{d\hat{I}_s}{dt_s} - \delta(t_f - t_0)(\hat{\mathcal{R}}_t - 1)\hat{I}_s\right]^2\Bigg|_{\bar{t}_{s,i}} + \\
& \frac{1}{N_C}\sum_{i=1}^{N_C}\left[\hat{\Delta}_{H,s} - \frac{\delta C\hat{\sigma}_s}{C_H}\hat{I}_s\right]^2\Bigg|_{\bar{t}_{s,i}} + \\
& \frac{1}{N_C}\sum_{i=1}^{N_C}\left[\hat{\Delta}_{I,s} - \delta\hat{\mathcal{R}}_t\hat{I}_s\right]^2\Bigg|_{\bar{t}_{s,i}}.
\end{aligned}
\tag{32}
$$

Hence, with the joint approach we aim at minimizing the functional:

$$
\begin{aligned}
\mathcal{L}_{HI,\text{joint}}(\hat{\Delta}_{H,s}, \hat{\Delta}_{I,s}, \hat{I}_s, \hat{\sigma}_s, \hat{\mathcal{R}}_t) \quad = \quad & \mathcal{L}_I(\hat{\Delta}_{I,s}) + \mathcal{L}_H(\hat{\Delta}_{H,s}) + \\
& \mathcal{L}_{HI,ODE}(\hat{\Delta}_{H,s}, \hat{\Delta}_{I,s}, \hat{I}_s, \hat{\sigma}_s, \hat{\mathcal{R}}_t).
\end{aligned}
\tag{33}
$$

By distinction, with the split approach we first train $\hat{\Delta}_{H,s}$ by the available data (see Eq (23)). Then, we use Eq (26) for $\hat{I}_s$ and:

$$\hat{\Delta}_{I,s} = \frac{C_H}{C} \frac{\hat{\mathcal{R}}_t \hat{\Delta}_{H,s}}{\hat{\sigma}_s} \tag{34}$$

for $\hat{\Delta}_{I,s}$, and minimize the functional:

$$
\begin{aligned}
\mathcal{L}_{HI,\text{split}}(\hat{\sigma}_s, \hat{\mathcal{R}}_t) = \quad & \frac{1}{N_D} \sum_{j=1}^{N_D} \left[ \frac{C_H}{C} \frac{\hat{\mathcal{R}}_t \hat{\Delta}_{H,s}(\tilde{t}_{s,j})}{\hat{\sigma}_s(\tilde{t}_{s,j})} - \tilde{\Delta}_{I,s,j} \right]^2 + \\
& \frac{1}{N_C} \sum_{i=1}^{N_C} \left\{ \frac{C_H}{C} \left[ \frac{d}{dt_s}\left( \frac{\hat{\Delta}_{H,s}}{\delta\hat{\sigma}_s} \right) - (t_f - t_0)(\hat{\mathcal{R}}_t - 1)\frac{\hat{\Delta}_{H,s}}{\hat{\sigma}_s} \right] \right\}^2 \Bigg|_{\tilde{t}_{s,i}}.
\end{aligned}
\tag{35}
$$

This choice for the split approach is based on the fact that hospitalization data are usually more reliable than infected individuals, hence they are more appropriate for an only-data regression training.

## 2.5 Simulation setup

The PINN-based approaches are here implemented by making use of the SciANN software library [33], a Keras and TensorFlow wrapper specifically developed for physics-informed deep learning. We analyze the performance of the PINN-based approaches to estimate the state variables and identify the governing parameters of an epidemiological model mimicking the setup of the first 90 days of a COVID-like disease outbreak in Italy. The total population is set to $N = 56 \times 10^6$ and the mean infectious period to $D = 5$ days, which is an estimate used for COVID-19 [4]. The initial value of infectious individuals $I_0$ is set to 1. The accuracy of the trained NNs is evaluated by the 2-norm of the error with respect to the 2-norm of the reference solution:

$$e_r = \frac{\| \hat{y} - y_{ref} \|_2}{\| y_{ref} \|_2}, \tag{36}$$

where $y$ can be either one of the state variables, or a time-dependent parameter. The relative error (36) is numerically computed by using 90 points equally spaced in the domain. We consider a number of scenarios, summarized in Table 1, differing for the reference SIR model and state variables of interest, the selection of the estimated governing parameters, and the available training data. The first five scenarios are used to validate the numerical model, while the last two consist of a real application to the Italian COVID-19 epidemic.

For the estimation of the transmission rate $\beta(t)$ in the basic SIR model (1) and (2), we consider three different scenarios:

- Case 1: constant $\beta$. We use this scenario to compare the efficiency of the joint and split approaches (10) and (11), respectively.

- Case 2: synthetic time-dependent $\beta(t)$, where the reference values are provided as an analytical function.

- Case 3: the reference $\beta(t)$ is obtained from the estimates of $\mathcal{R}_t$ in the first months of the COVID-19 epidemic outbreak in Italy.

**Table 1. Scenarios adopted to analyze the proposed PINN-based approaches.**

|  | State variables | Estimated parameters | Reference values | Training data |
|---|---|---|---|---|
| Case 1 | $S,I,R$ | $\beta$ (constant) | $\beta_0 = 0.6 \text{ d}^{-1}$ | $\tilde{I}_j$ (Poisson error) |
| Case 2 | $S,I,R$ | $\beta$ (time-dependent) | Synthetic $\beta$ | $\tilde{I}_j$ (Poisson error) |
| Case 3 | $S,I,R$ | $\beta$ (time-dependent) | COVID-19 $\mathcal{R}_t$ | $\tilde{I}_j$ (Poisson error) |
| Case 4 | $I$ | $\mathcal{R}_t$ | Synthetic $\beta$ | $\tilde{I}_j$ (40% Gaussian error) |
| Case 5 | $I,\Delta_H$ | $\mathcal{R}_t,\sigma$ (time-dependent) | Synthetic $\beta,\sigma$ | $\tilde{I}_j$ (40% Gaussian error), $\tilde{\Delta}_{Hj}$ (Poisson error) |
| Case 6 | $\Delta_I,\Delta_H$ | $\mathcal{R}_t,\sigma$ (constant) | COVID-19 $\mathcal{R}_t$ | $\tilde{\Delta}_{Ij}, \tilde{\Delta}_{Hj}$ (COVID-19 dataset) |
| Case 7 | $\Delta_I,\Delta_H$ | $\mathcal{R}_t,\sigma$ (time-dependent) | COVID-19 $\mathcal{R}_t$ | $\tilde{\Delta}_{Ij}, \tilde{\Delta}_{Hj}$ (COVID-19 dataset) |

In each case, the training data are the number of infectious individuals per day. These are synthetically generated by numerically integrating the system (1) using the selected reference function for $\beta(t)$. In particular, we used $N_D = 90$ training data points (one value per day). To take into account possible reporting errors, the data $\tilde{I}_j$ for each time $\tilde{t}_j$ are obtained by sampling from a Poisson distribution having as mean $I(\tilde{t}_j)$. This kind of Poisson error is frequently assumed on data arising from a counting process. The infectious data and the epidemiological model are scaled by a factor $C = 10^5$ (see Eq (5)).

The reduced SIR model (12) is then used to explore a more realistic scenario with strongly perturbed data on the infected individuals and accurate data on the number of hospitalizations. The joint and split approaches (Eqs (15) and (16)) are used to estimate the governing parameter $\mathcal{R}_t$ in the following inverse problems:

- Case 4: synthetic time-dependent $\beta(t)$ (as in Case 2), subject to a larger error noise on the infectious data.

- Case 5: an adaptation of Case 4, considering also the hospitalization data and a time-dependent hospitalization fraction $\sigma$ (to be estimated).

In Case 5, the synthetic data of the daily hospitalizations, $\{\tilde{\Delta}_{Hj}\}_{j=1}^{N_D}$, are obtained by sampling from a Poisson distribution having as mean value the reference solution. The scaled values are obtained by setting $C_H = 10^3$.

Finally, we applied the PINN approaches to the infected and hospitalized data reported in Italy during the first months of the COVID-19 pandemic:

- Case 6: infers a time-dependent $\mathcal{R}_t$ while considering $\sigma$ as a constant.

- Case 7: simultaneously infers $\mathcal{R}_t$ and $\sigma$ as functions of time.

In these scenarios we consider the epidemiological data provided by the Italian surveillance system [34] from February 21st, 2020 to May 20th, 2020. The period coincides with the advent of the disease and its initial spread. The vaccination campaign was not started yet and possible reinfections are negligible. The Italian dataset contains the number of new daily hospitalizations and reported infections, $\{\tilde{\Delta}_{Hj}\}_{j=1}^{N_D}$ and $\{\tilde{\Delta}_{Ij}\}_{j=1}^{N_D}$, respectively, and supplies an estimate of the COVID-19 reproduction number $\mathcal{R}_t$ based on [10]. The scaled values are obtained by setting $C$ and $C_H$ equal to the maximum experimented values for $\Delta_I$ and $\Delta_H$, respectively, in the 90 days taken into consideration. New infections are multiplied by a reporting ratio $\alpha_r = 6$, following the estimate from Italian Institute of Statistic based on the sierological data [35].

## 2.6 Implementation details

The best architecture of the NNs is typically dependent on the desired application. On the one hand, the number of neurons and hidden layers should be large enough to make the PINN-based model able to reconstruct the epidemic dynamics. On the other hand, parsimonious NNs are required to contain the number of parameters, limit the computational times of the training process, and avoid overfitting.

To select an adequate architecture for the considered PINN model, we compared the error, the training times and the number of parameters obtained in Cases 1 and 2 using different number of layers (4, 10) and neurons (5, 25, 50, and 100). The details about this sensitivity analysis are reported in Appendix A in S1 Text. Neural Network architectures. Hence, the NNs for $\hat{S}_s$ and $\hat{I}_s$ are built with 4 hidden layers, 50 neurons for each and tanh as activation function. In Case 1 the constant transmission rate is treated as a single parameter in the training. In Cases 2 and 3, $\hat{\beta}_s$ has 4 hidden layers with 100 neurons. In Cases 4 and 5 the NN for $\hat{\mathcal{R}}_t$ has the same architecture described for $\hat{\beta}_s$. In Cases 6 and 7 the NN for $\hat{\mathcal{R}}_t$ has 4 hidden layers with 100 neurons each. In Cases 5 and 7 the NN for $\hat{\sigma}_s$ has 10 hidden layers with 5 neurons each. In all scenarios we consider $N_C$ = 6000 collocation points randomly sampled in $[t_0, t_f]$ from a uniform distribution. The NNs are initialized using Glorot initialization [36] and trained using Adam optimization algorithm [37] with a reduced-on-plateau learning rate schedule, which is initialized to 0.001 and halved if learning stagnates.

In the joint approach we trained the NNs for 5000 epochs with batches containing 100 training points. In the split approach we set the number of epochs to 3000 for the data fitting training and to 1000 for the fully-physics-informed regression, with a batch size equal to 10 and 100, respectively. When the total number of training points ($N_D + N_C$) is large, it is a good practice to use a mini-batch gradient descent as optimization algorithm, which splits the training dataset into small batches that are used to calculate the loss function and update the model coefficients at each epoch. The weights $\omega_*$ are calibrated in the fitting process using the eigenvalues of the Neural Tangent Kernel (NTK) [38], with the specific SciANN built-in function. The approach leverages the NTK to dynamically and adaptively tune the loss-term weights, enhancing the performance and robustness of PINNs in solving differential equations and other physics-informed tasks. Indeed, not using NTK adaptive weights results in larger errors as it is shown in detail in Appendix A in S1 Text. All simulations were performed on a machine with two Intel(R) Xeon(R) CPU E5–2680 v2 @ 2.80GHz and 256GB of RAM. We report the version of the main libraries that were used: Tensorflow = 2.5.3, Keras = 2.5.0, SciANN = 0.6.6.1.

# 3 Numerical results

## 3.1 Model validation

**3.1.1 Case 1: Constant transmission rate.** The first scenario assumes a constant transmission rate during the simulation. The reference value for the parameter is fixed to $\beta = \beta_0 = 0.6$ $d^{-1}$. The corresponding basic reproduction number is $\mathcal{R}_0 = 3$, which is an estimate of the basic reproduction number in the COVID-19 epidemic in Italy. The resulting system dynamic is shown in Fig 3a. The evaluation of a constant parameter does not require an additional NN for $\beta$, and it is implemented through an object of the SciANN parameter class with the additional non-negative constraint. Both the joint and split approaches obtain solutions that match well the reference dynamic (see Fig 3a for the results of the split approach). Fig 3b shows that the convergence of the joint approach to the reference value is slower than the split approach.

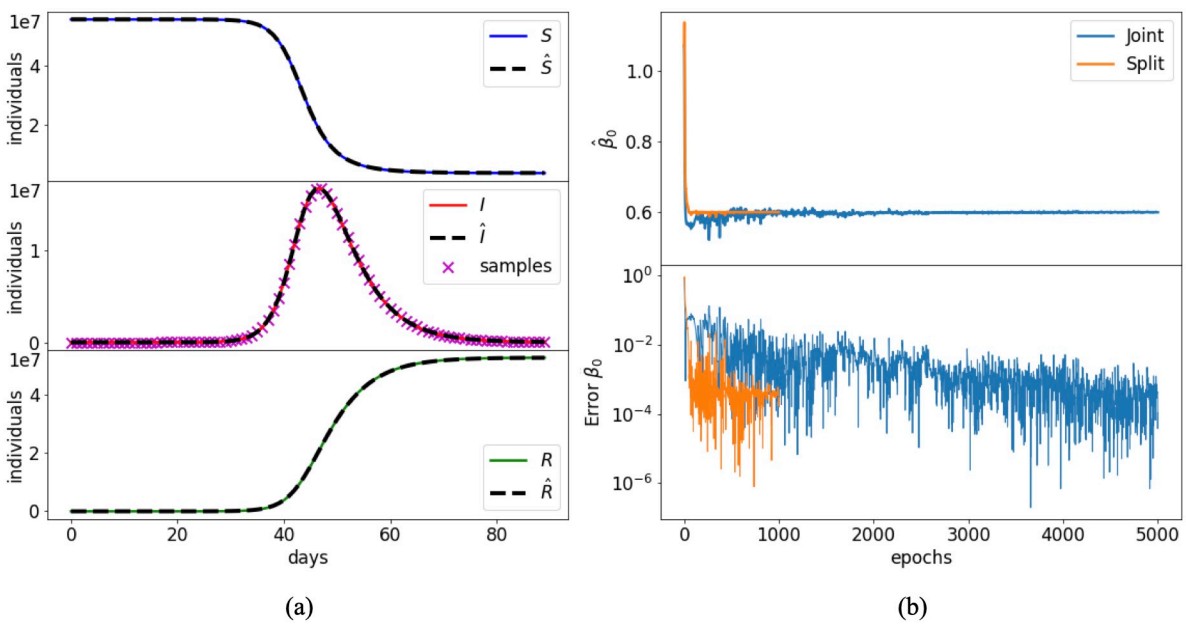

**Fig 3. Case 1: Constant transmission rate.** Comparison between (a) reference solutions of the SIR model (1) and PINN approximations in the split approach; (b) $\beta_0$ identification during training of the joint and the split methods.

In fact, there are strong oscillations during training. The split approach mitigates these oscillations and reaches a faster convergence to the reference value.

Table 2 provides a quantitative comparison of the resulting errors for each state variable along with the training times. Note that for the split approach the sum of the only-data and physics-informed training is reported. In both approaches the PINN solution reaches an acceptable accuracy, with a relative error on the order of $10^{-3}$ for all state variables, resulting in a good estimate of the unknown parameter $\beta$ as well. The split approach reduces the total training time about by a factor 3, and improves the accuracy with respect to the joint approach by one order of magnitude on average, hence it appears to be in this case largely preferable.

A second test is carried out by changing the amount of available data for the training. For instance, we consider only weekly values for the infection data, which are more likely to compensate the errors and oscillations of daily data. This reduces the number of training points to $N_D = 13$. The model is built and trained as stated in Section 3, with the difference that the training of $\hat{I}_s$ in the split method is performed at each epoch on the whole data set and the

**Table 2. Case 1: Constant transmission rate.** Training time, approximation errors, and estimations of the joint and split methods for $\beta = 0.6$ d$^{-1}$. The daily and weekly infection data correspond to $N_D = 90$ and $N_D = 13$, respectively.

| | Daily data | | Weekly data | |
|---|---|---|---|---|
| | **Joint** | **Split** | **Joint** | **Split** |
| Training time [s] | 2223 | 719 | 1977 | 493 |
| Error $S$ | $2.763 \times 10^{-3}$ | $6.221 \times 10^{-4}$ | $2.833 \times 10^{-2}$ | $2.247 \times 10^{-2}$ |
| Error $I$ | $3.846 \times 10^{-3}$ | $8.444 \times 10^{-4}$ | $4.544 \times 10^{-2}$ | $4.245 \times 10^{-2}$ |
| Error $R$ | $3.258 \times 10^{-3}$ | $7.434 \times 10^{-4}$ | $3.276 \times 10^{-2}$ | $2.491 \times 10^{-2}$ |
| Error $\beta$ | $5.086 \times 10^{-3}$ | $3.587 \times 10^{-4}$ | $4.693 \times 10^{-2}$ | $5.187 \times 10^{-2}$ |
| PINN $\hat{\beta}$ | 0.59738 | 0.60007 | 0.57323 | 0.56888 |

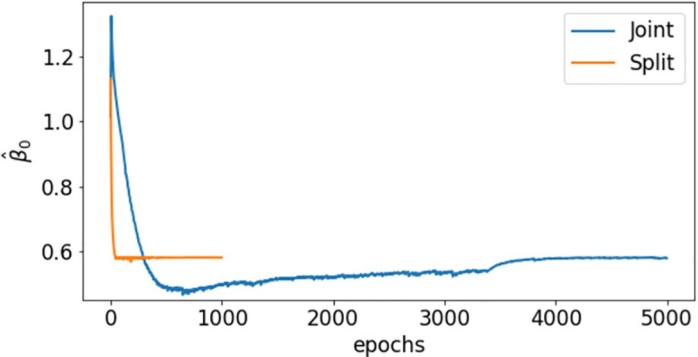

**Fig 4. Case 1: Constant transmission rate.** Comparison between $\beta_0$ identification during the training of the joint and split methods for the weekly infection data ($N_D = 13$).

mini-batches are not needed. Moreover, given the overall small size of the training dataset, we can decrease the maximum number of epochs in the data regression from 3000 to 1000. Training times and approximation errors reported in Table 2 show an equivalent outcome in terms of accuracy, but with a strong gain in training time for the split approach. By reducing the number of samples the amount of information is smaller, hence the accuracy decreases with respect to a daily updated information. The convergence of the parameter $\beta$ has a similar behavior to the one shown previously (Fig 4), with the split method reducing both the oscillations and the convergence time.

**3.1.2 Cases 2 and 3: Time-dependent transmission rates.** Case 2 consists of a simulation of disease spread according to the time-dependent transmission rate plotted in Fig 5. This $\beta(t)$ behavior implies two waves of infection, with a maximum number of infectious individuals two orders of magnitude lower than in Case 1.

Fig 5 shows the reference values of the unknowns and the corresponding NN approximations. The dashed lines correspond to the mean outcome from 10 different runs, while the grey bands provide the confidence interval of plus/minus a standard deviation.

At most times the split method provides more stable and accurate results, with smaller variations from one run to another. The higher stability and speed of convergence of the split approach can be also appreciated from the error behavior on $\hat{\beta}_s$ during the training (Fig 6). The overall performance of the PINN approaches is summarized in Table 3.

Both approaches, however, fail to estimate the initial values of $\beta$ (Fig 5). The low number of infected individuals and the presence of perturbed data produce an initial dynamic that the NNs erroneously learn by considering a larger number of initial infected individuals and a lower initial transmission rate. Errors of this kind represent a common hurdle in epidemiology, as the evolution of a disease is extremely difficult to be identified at the beginning of the epidemic outbreak under the assumption of a temporal-depending transmission rate. For this reason, Table 3 provides also the error for $\beta$ during the last 70 days of simulation. These small errors confirm the accuracy of the PINN estimates when the data provide a clear signal.

Case 3 considers the $\mathcal{R}_t$ estimates supplied by the Italian health institute "Istituto Superiore della Sanità" (ISS) [34] as reference values for the effective reproduction number. These values, depicted in Fig 7, are used to evaluate the associated transmission rate $\beta$ in the model and to synthetically generate the epidemiological data of infected individuals from February 21st to May 20th, 2020.

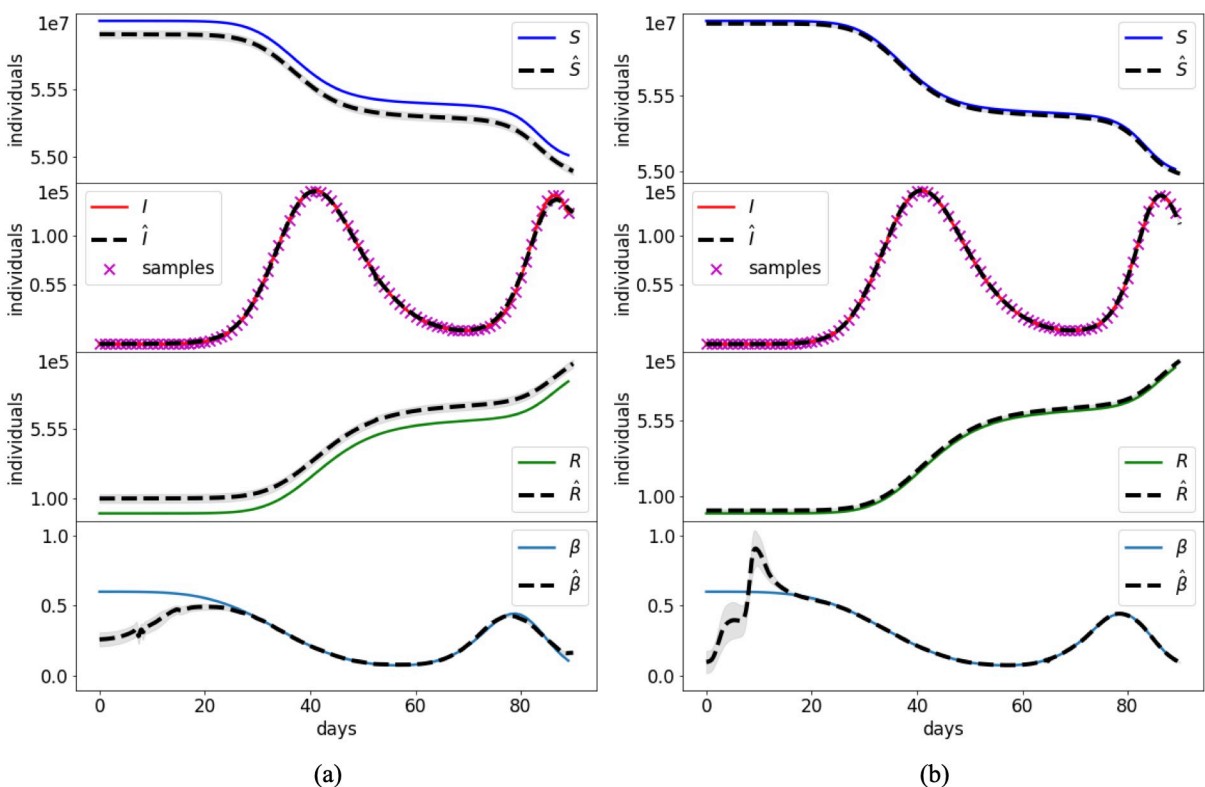

**Fig 5. Case 2: Synthetic transmission rate.** Comparison between the reference solution of the SIR model and the PINN approximations with the joint (a) and split (b) approach. Grey bands provide the confidence interval of one standard deviation.

Note that for the first 20 days the values have been kept equal to 3.012, in order to simulate the free transmission of the pathogen in a completely susceptible population without restrictions in contacts. Fig 7 and Table 3 contain the outcome of the joint and split methods. Similarly to Case 2, the estimation of the temporal evolution of the transmission rate is hard for the initial times. Both methods can achieve a good accuracy after the first 20 days, as it can be deduced from Fig 7 and from the errors associated to the last 70 days of the simulation. The

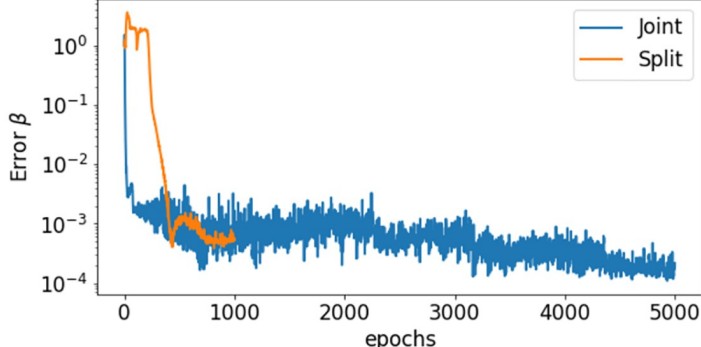

**Fig 6. Case 2: Synthetic transmission rate.** Comparison between the errors on $\hat{\beta}_s(t)$ during the training with the joint and split approach.

**Table 3. Cases 2–3: Time-dependent transmission rates.** Training time and approximation errors for the state variables and the estimated parameters with the joint and split approach.

| | Joint | Split |
|---|---|---|
| **Case 2** | | |
| Training time [s] | 1936 | 717 |
| Error $S$ | $1.814 \times 10^{-3}$ | $3.381 \times 10^{-4}$ |
| Error $I$ | $2.501 \times 10^{-2}$ | $7.754 \times 10^{-3}$ |
| Error $R$ | $2.288 \times 10^{-1}$ | $4.251 \times 10^{-2}$ |
| Error $\beta$ | $2.678 \times 10^{-1}$ | $4.127 \times 10^{-1}$ |
| Error $\beta$ (last 70d) | $5.979 \times 10^{-2}$ | $1.563 \times 10^{-2}$ |
| **Case 3** | | |
| Training time [s] | 1906 | 726 |
| Error $S$ | $1.935 \times 10^{-3}$ | $5.031 \times 10^{-4}$ |
| Error $I$ | $4.729 \times 10^{-3}$ | $6.006 \times 10^{-3}$ |
| Error $R$ | $2.805 \times 10^{-2}$ | $7.101 \times 10^{-3}$ |
| Error $\mathcal{R}_t$ | $4.692 \times 10^{-1}$ | $6.988 \times 10^{-1}$ |
| Error $\mathcal{R}_t$ (last 70d) | $5.341 \times 10^{-2}$ | $6.701 \times 10^{-2}$ |

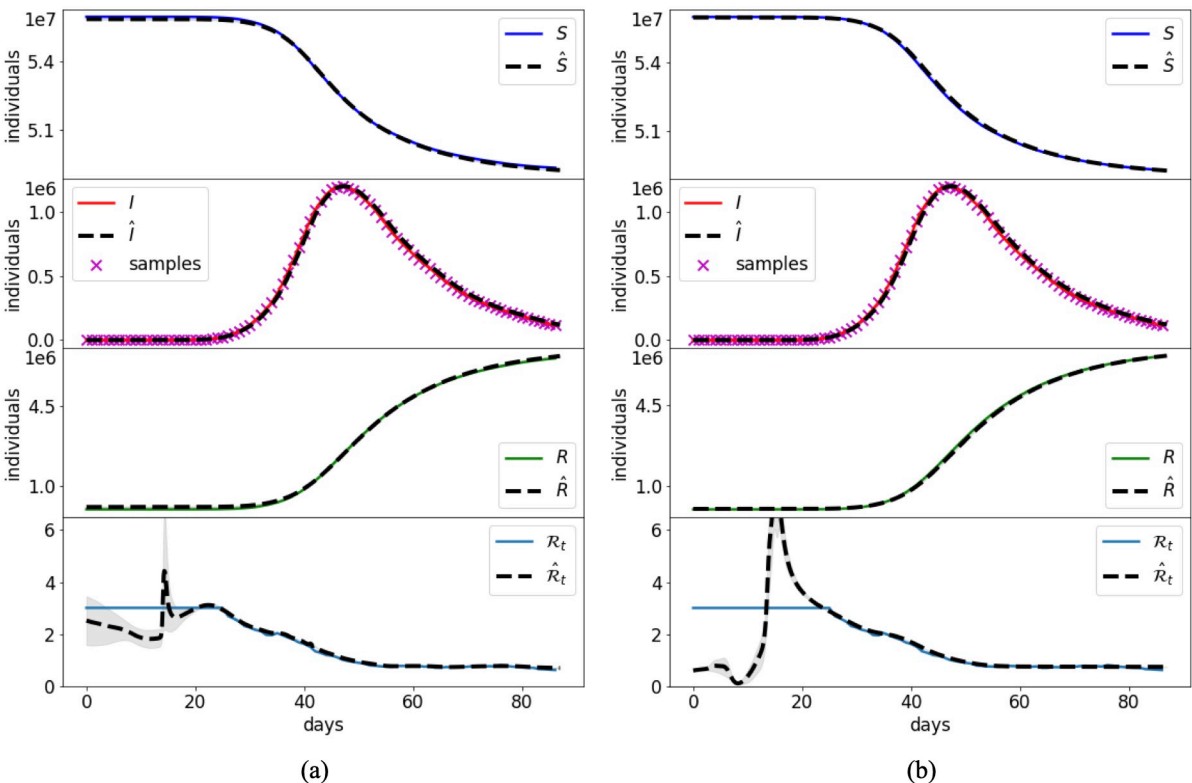

(a)                                              (b)

**Fig 7. Case 3: Transmission rate of the Italian COVID-19.** Comparison between the reference solution of the SIR model and the PINN approximations with the joint (a) and split (b) approach. Grey bands provide the confidence interval for one standard deviation.

matching is quite accurate, with relative errors on the order of $10^{-3}$, and the variance from 10 different runs is also limited. While the gain in accuracy of the split method is not so clear in this case, the benefit in the training duration is still relevant. Similar results in terms of accuracy have been obtained on these test cases when using the reduced implementation (12) of the split and joint approaches.

**3.1.3 Cases 4 and 5: Reduced model and hospitalization data.** Case 4 is used to investigate the performance of the joint and split approaches for the reduced model (12) in a synthetic scenario where the data are subject to large errors. The unknown time-dependent transmission rate is the same as the one of Case 2, whose corresponding reproduction number is shown in Fig 8.

The time domain is extended to $t_f = 120$ days, which implies two complete waves of infection. To simulate the typically large uncertainties on the reported daily infections, we generate synthetic noisy data by perturbing the numerical solution for $I(\tilde{t}_j)$ with a Gaussian error having zero mean and coefficient of variation of 40%. The data are then rounded to the closest integer with the negative values set to 0.

The outcome of the PINN approximations are provided in Fig 8 and Table 4. The larger errors on the data reduce the accuracy in the estimate of the reproduction number. The two approaches are almost equivalent in terms of accuracy, but the split one has faster training times. Clearly, a large uncertainty in the training data reduces the reliability of the split approach, which mostly relies on this piece of information, while the joint approach compensate possible errors on data with the residuals of the governing equations. Nevertheless, the accuracy in the reproduction of the $\mathcal{R}_t$ appears to be in any case fairly satisfactory, given the large reported errors. It is worth noting that the solution with PINNs of the proposed reduced model clearly outperforms the one of the full SIR model when the data are subject to large errors, as reported in Appendix B in S1 Text.

Case 5 aims at improving the estimate of $\mathcal{R}_t$ by introducing the information on the daily hospitalizations. This entails that also the unknown parameter $\sigma$, which is fraction of infected individuals that become hospitalized, becomes a NN to be trained. The reference $\sigma$ for this case is shown in Fig 9. The results of the joint and split approaches are summarized in Fig 9 and Table 4. The inclusion of more reliable data implies a better estimate of $\mathcal{R}_t$ with respect to

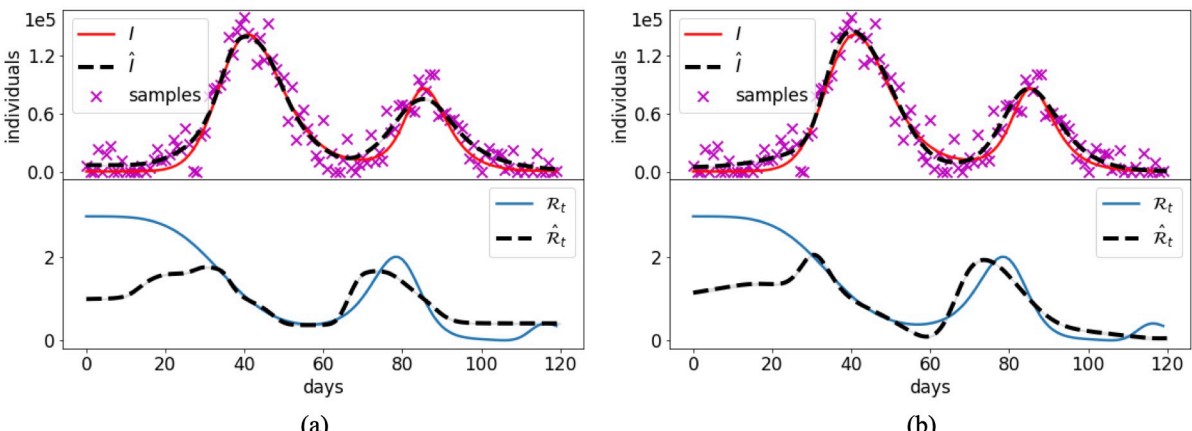

**Fig 8. Case 4: Synthetic transmission rate with large data errors.** Reference solution and PINN approximation with the joint (a) and split (b) approach in the reduced SIR model (12) and noisy data.

**Table 4. Cases 4–5: Reduced model and hospitalization data.** Training time and approximation errors for the state variables and the estimated parameters with the joint and split approaches when considering the hospitalization data (Case 5) or not (Case 4).

| | Joint | Split |
|---|---|---|
| **Case 4** | | |
| Training time [s] | 1658 | 1053 |
| Error $I$ | $1.411 \times 10^{-1}$ | $1.331 \times 10^{-1}$ |
| Error $\mathcal{R}_t$ | $4.961 \times 10^{-1}$ | $4.744 \times 10^{-1}$ |
| Error $\mathcal{R}_t$ (last 100d) | $3.141 \times 10^{-1}$ | $3.336 \times 10^{-1}$ |
| **Case 5** | | |
| Training time [s] | 1829 | 1445 |
| Error $\Delta_H$ | $8.783 \times 10^{-3}$ | $4.722 \times 10^{-3}$ |
| Error $I$ | $1.209 \times 10^{-1}$ | $7.434 \times 10^{-2}$ |
| Error $\mathcal{R}_t$ | $4.407 \times 10^{-1}$ | $4.992 \times 10^{-1}$ |
| Error $\mathcal{R}_t$ (last 100d) | $2.410 \times 10^{-1}$ | $1.240 \times 10^{-1}$ |
| Error $\sigma$ | $3.858 \times 10^{-1}$ | $1.417 \times 10^{-1}$ |
| Error $\sigma$ (last 100d) | $2.626 \times 10^{-1}$ | $1.145 \times 10^{-1}$ |

Case 4. This is particularly evident for the split approach, which is able to provide good estimates of both temporal depending parameters ($\mathcal{R}_t$ and $\sigma$), and thus, lower errors (Table 4).

## 3.2 Application to real data

### 3.2.1 Cases 6 and 7: Italian COVID-19 surveillance data.

Cases 6 and 7 apply the procedure adopted in Case 5 to the real setting of the Italian COVID-19 epidemic outbreak. The fraction of the number of infected individuals that becomes hospitalized, $\sigma$, is assumed to be

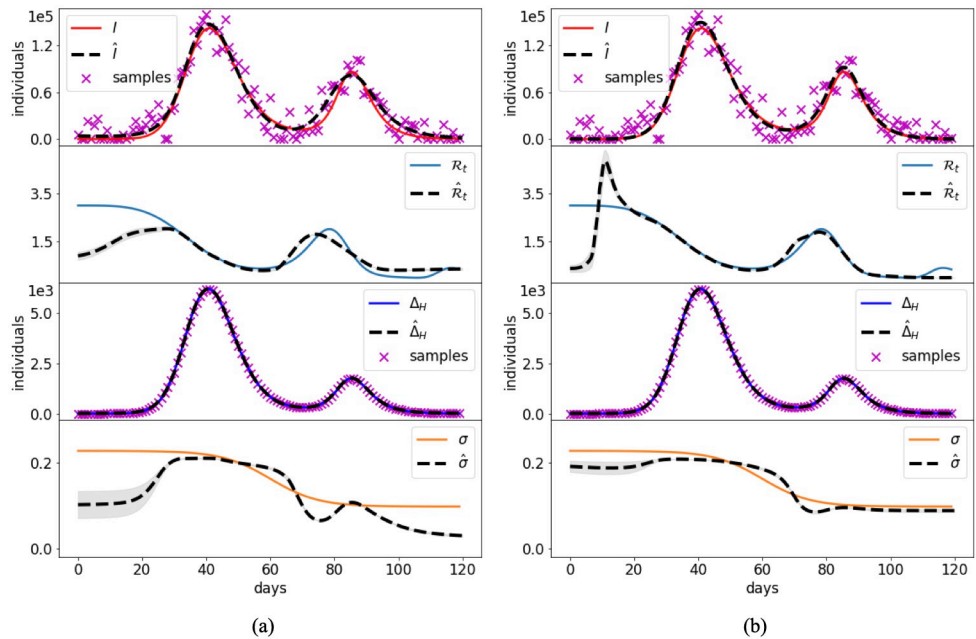

(a)

(b)

**Fig 9. Case 5: Synthetic data of infections and hospitalizations.** Reference solution and PINN approximation in the joint (a) and split (b) approaches.

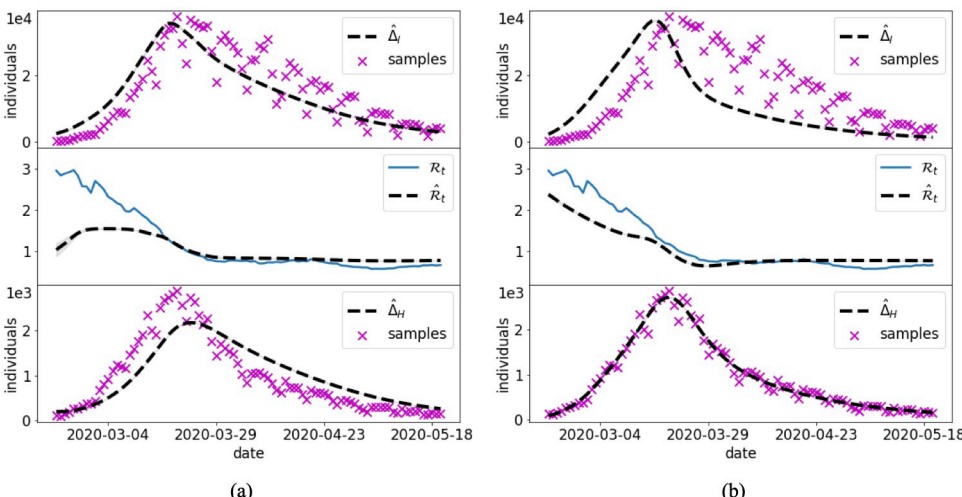

(a)                                                     (b)

**Fig 10. Case 6: Constant hospitalization rate.** PINN approximations with the joint (a) and split (b) methods for the Italian COVID-19 outbreak with constant $\sigma$. The blue trajectory of $\mathcal{R}_t$ is the official estimate by ISS.

constant in time in Case 6 and time-dependent in Case 7. The goal is to estimate $\mathcal{R}_t$ and $\sigma$ by means of the presented PINN approaches.

The results of the joint and the split approaches are shown in Figs 10 and 11 for Cases 6 and 7, respectively. We use the reproduction number evaluated by ISS (Fig 10) as a reference value for comparison. However, it is important to keep in mind that its values have been obtained with a data driven approach (renewal equation, [10]) on the symptomatic infected individuals. In Case 6 the joint approach provides an acceptable accuracy on the new infection data, while

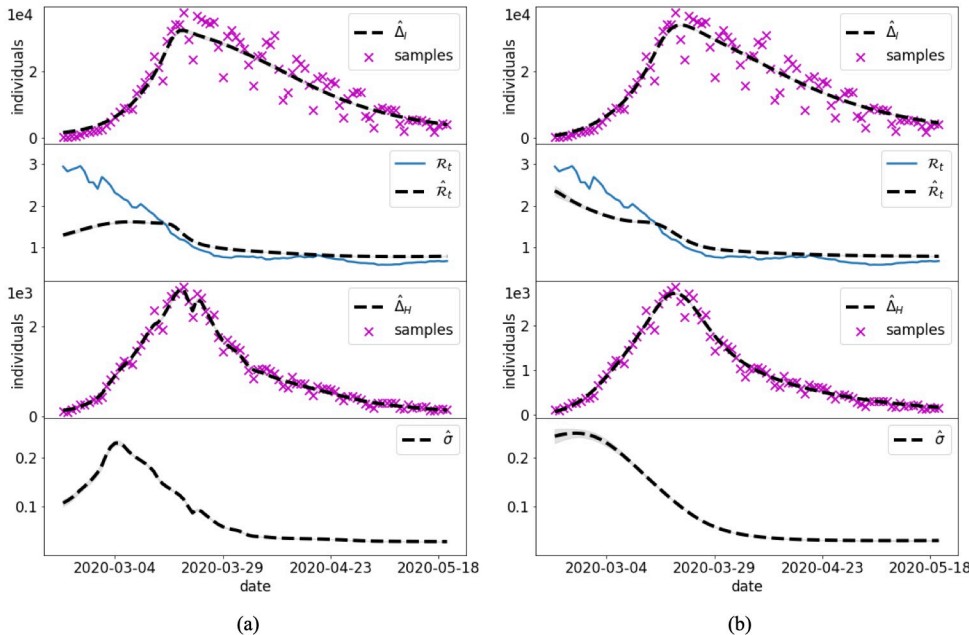

(a)                                                     (b)

**Fig 11. Case 7: Time-dependent hospitalization rate.** PINN approximations with the joint (a) and split (b) methods for the Italian COVID-19 outbreak with time-dependent $\sigma$. The blue trajectory of $\mathcal{R}_t$ is the official estimate by ISS.

the peak of the daily hospitalizations is underestimated (Fig 10a). The split approach, instead, is firstly trained on the daily hospitalization well retrieving these high-fidelity data. The second part of the training, considering both the daily infections and the reduced model equations, does not achieve the same level of accuracy for the infection data with respect to the joint approach (Fig 10b). For what concerns the estimation of $\mathcal{R}_t$, the joint approach strongly underestimates its value at the beginning of the epidemic. By distinction, the split approach provides a trend of $\mathcal{R}_t$ that is fully consistent with the ISS estimates, i.e., a decrease during the first weeks of the epidemic, followed by an almost stationary value around 1 during the recession phase. The overall performance of the PINN approaches are summarized in Table 5.

Considering a time-dependent $\sigma$ (Case 7) helped both approaches to improve the estimate of both hospitalized and infectious data. Both approaches depict a similar trend for $\sigma$, where the fraction of infected individuals that became hospitalized decreases during March 2020 from a peak of about 23% to about 3%, which is a realistic outcome in the framework of Italian COVID-19 outbreak. The main differences among the two approaches are still at the beginning of the outbreak, where the joint approach suggests a lower value of $\sigma$, while underestimating the values of $\mathcal{R}_t$. The training time required by the split approach was about 40% of the time for the joint approach (Table 5).

**3.2.2 Forecasting.**   Cases 1–7 use the complete set of data to infer the past dynamics of the unknown state variables and parameters. This section analyzes the performance of the PINN methods in a realistic forecasting scenarios where PINNs are trained using only a portion of the data, and then employed to produce a short-term forecast of the epidemic. TNote that the training still consists of an inverse problem, since the time-dependent parameters of the model are still unknown.

In the Italian COVID-19 setting of Case 7, we trained the PINN model on four temporal windows of increasing length. As first, the PINN model is trained on the data from February 21st, 2020 (day 0) to March 6th, 2020 (day 15). The NNs are further trained on the data of the other windows (days [0, 30], [0, 45], [0, 60]), i.e. the training of the same architectures is carried on by sequentially adding new data corresponding to longer temporal widows. This approach has the advantage of improving the previously trained NNs, instead of starting new NNs from scratch. Short term forecasts of 15 days are produced at the end of each training window using the neural networks in extrapolation. Fig 12 compares the trained solutions and the forecasts at each window with the reported data. The PINN solution obtained by training the NNs on the whole set of data (result of Case 7) is shown as reference solution.

As one could expect, the projections are not accurate when few data are available or when the model is close to the peak of the epidemic (Fig 12a and 12b). Good outcomes are achieved during the recession (Fig 12c and 12f).

**Table 5. Cases 6–7: Application to the Italian COVID-19 data.** Training time and approximation errors for the state variables and the estimated parameters with the joint and split approach.

|  | Joint | Split |
| --- | --- | --- |
| **Case 6** |  |  |
| Training time [s] | 1655 | 1042 |
| Error $\mathcal{R}_t$ | $3.881 \times 10^{-1}$ | $2.495 \times 10^{-1}$ |
| Estimated $\hat{\sigma}$ | 0.0686 | 0.0849 |
| **Case 7** |  |  |
| Training time [s] | 1778 | 1101 |
| Error $\mathcal{R}_t$ | $3.646 \times 10^{-1}$ | $2.236 \times 10^{-1}$ |

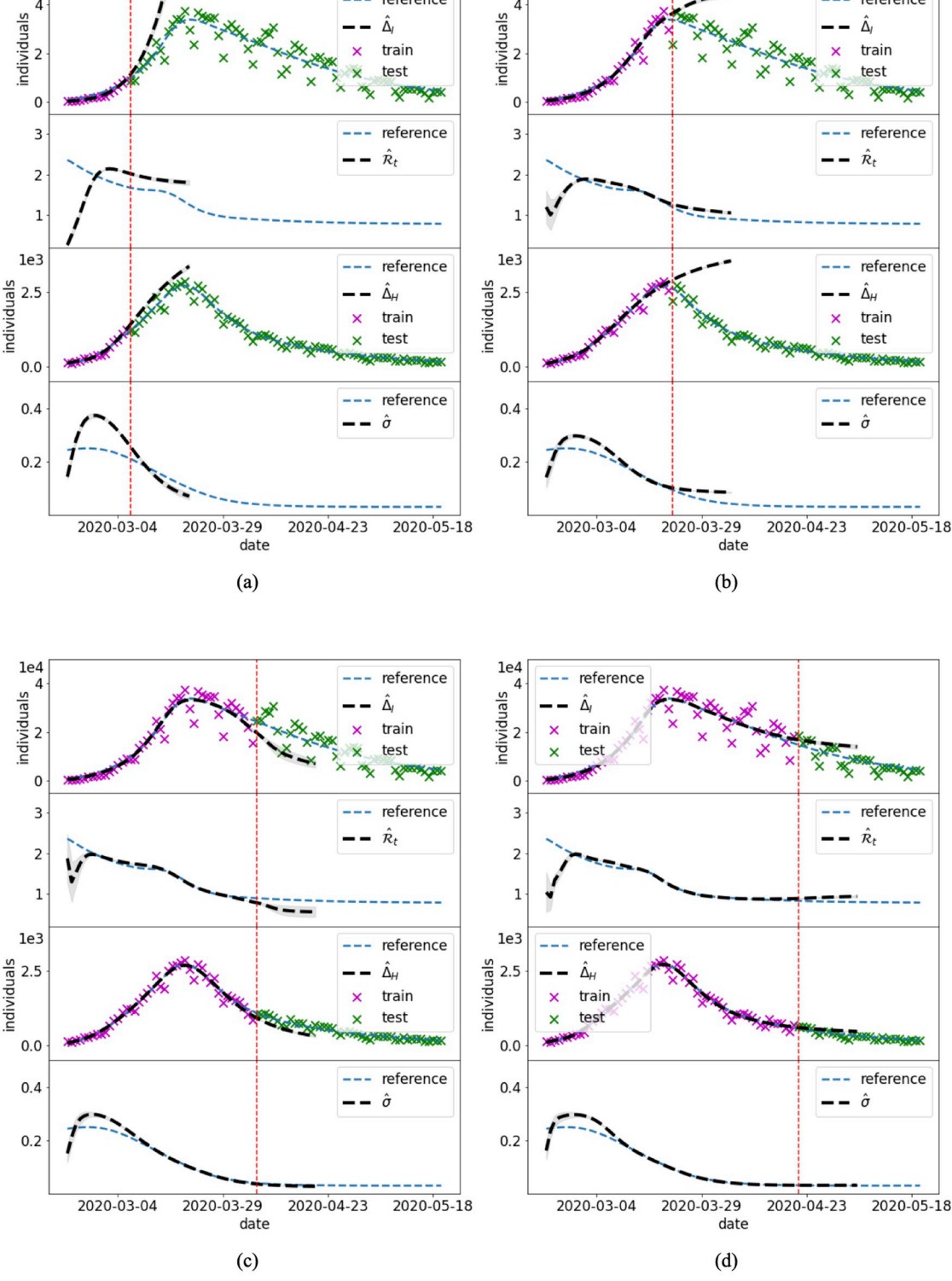

**Fig 12. Case 7: Forecasting.** PINN predictions of the Italian COVID-19 evolution using the split method on subsequent training time intervals: (a) 0–15 days, (b) 0–30 days, (c) 0–45 days, (d) 0–60 days. The blue dashed lines, here used as reference solution, are the outputs of the split approach computed in Case 7 (Fig 11b).

Similar forecast results are obtained using the joint PINN approach (see Appendix C in S1 Text).

## 4 Discussion and conclusions

This work proposes two innovative ideas to improve the application of PINNs for the solution of SIR-based epidemiological models, and to estimate the time-dependent transmission rate, or the effective reproduction number, of an epidemic. The first idea consists in splitting the training of the NNs in two steps: the first step provides the fit on the epidemiological data, while the second step minimizes the residual on the model equations. The performance of the split approach has been compared to a standard PINN application, which trains simultaneously the NNs on the joint loss function on data and residual. The second idea consists in implementing a modification to the basic model equations, possibly removing the state variables that are not directly related in the disease transmission and the associated redundant terms in the loss function. This reduced PINN model has been extended to include both infection cases and hospitalization data, which are usually more reliable pieces of information.

Synthetic test Cases 1–3 showed that, when infectious data are subject to small errors, both the split and joint PINN approaches are able to retrieve with high accuracy the system dynamics. The initial training on the data of the split approach provides a clear advantage when minimizing the residual on the model equations and estimating the reproduction number. In fact, it achieves lower errors (up to an order of magnitude) with faster computational times (speed up larger than 60% in Cases 2 and 3). This is probably due to more stable results during the training epochs, as depicted in Fig 3b for a constant transmission (Case 1), and Fig 6 for a time-dependent transmission (Case 2). However, the simultaneous estimate of the initial conditions and the initial transmission proved to be particularly challenging for both PINN approaches (Figs 5 and 7). In fact, even small errors on the data become particularly relevant when there are low number of infections, such as at the beginning of the epidemic. Model results could improve by assuming a constant initial transmission rate, which is generally in agreement with the free circulation of the pathogen in absence of interventions.

Besides this inaccuracies in the early times of the outbreak, the $\mathcal{R}_t$ estimated by PINN in Case 3 has a similarly accuracy to the one obtained with the renewal equation [10], approach that is commonly adopted during outbreaks (see Appendix D in S1 Text).

The large errors that typically characterize the data of the reported daily infections might deteriorate the retrieval of the temporal changes of the transmission rate and the associated effective reproduction number (Case 4, Fig 8 and G in S1 Text). For this reason, numerous epidemiological analysis are based on data that are less biased by the surveillance system, such as daily hospitalization data, e.g., [4]. Case 5 shows that the use of daily hospitalizations and infections into the reduced PINN model allows to improve the accuracy in the estimation of the uncertain time-varying parameters, in particular both the effective reproduction number $\mathcal{R}_t$ and the fraction of infected individuals requiring hospitalization $\sigma$. The split approach still outperforms the joint counterpart with 20% savings in the training cost, however both approaches still present limitations at the beginning of the outbreak.

The application to the Italian COVID-19 epidemic (Cases 6 and 7) emphasizes the importance of considering the fraction of hospitalized individuals, $\sigma$, as a temporal-dependent parameter. In fact, the PINN approximations were not able to accurately follow both time series of daily hospitalized and infectious data when considering a constant $\sigma$ (Fig 10). Results notably improve in the case of a temporal-dependent $\sigma$ (Fig 11). Many modeling studies assume a constant $\sigma$, with possible temporal variations assigned only on the arrival of new disease variants or after the deployment of vaccines. Other processes that might directly impact

this parameter are typically neglected. For example, in the early stages of the outbreak, the fear of the new disease might prompt many symptomatic infected individuals to seek health care at the hospital (thus generating a large value of $\sigma$). The subsequent overcrowding of the hospitals and improvement of treatment at home might reduce the value of $\sigma$ in time. The time-dependent $\sigma$ values estimated by the PINN approaches (Fig 11) show exactly such a dynamic, with small differences at the beginning. We argue that also in this application the split approach outperforms the joint one. Besides the advantages in the computational times (40% faster), the effective reproduction number obtained with the split approach depicts a closer trajectory to the reference $\mathcal{R}_t$ estimated by the Italian Institute of Health (Fig 11).

The shorter training times and higher accuracy that we consistently obtained for the split approach in each test case can be attributed to the structure of the loss functions. Splitting the training implies the minimization of two loss functions which are simpler, thus sparing the complex solution of a multi-objective optimization problem.

The results presented in Cases 1–7 demonstrate the ability of the PINNs model to retrieve the past dynamics of an epidemic, to infer the temporal changes in the parameters, and to possibly fill the gaps among the data (Fig 4). A clear benefit of PINNs with respect to other traditional epidemiological approaches is that PINNs can directly produce short-term forecasts and projections of the epidemics. In fact, the calibrated PINNs are functions of time and can be extrapolated outside the training window. This is explored by analyzing the forecasts produced by the split PINN approach in the framework of the Italian COVID-19 pandemic (Fig 12).

In general, forecasts based on extrapolation might be extremely far from the real data due to possible strong fluctuations of the functions outside the training window. Our results show that the PINNs show smooth behaviors also in extrapolation, which in general are consistent with the dynamics of the disease spread. We attribute this consistency to the fact that the NNs are trained using the residual of the ODEs, which informs the model of the ongoing trend. PINNs, thus, can be particularly interesting for short-term forecasting as well. As one could expect, the quality of the forecast changes for different training window. If the training interval stops in conjunction with a peak of the cases, the model hardly predicts the fall in infections in the following days, as in Fig 12b. Instead, predictions during recession are more accurate (Fig 12c-d). This kind of results are common to many models that attempt predicting the dynamics of an outbreak. Improvements could be achieved, for example, by using universal models trained on scaled data [39].

A major limitation of the PINN approach with respect to traditional statistical methods for $\mathcal{R}_t$ inference (such as [10]), is that the deterministic nature of PINN does not provide a quantification of the uncertainty. For example, when dealing with strongly perturbed data as in Case 4, it would be reasonable to expect a large confidence interval around the estimated $\mathcal{R}_t$ values (see Fig G in S1 Text). Uncertainty quantification (UQ) is a fundamental ingredient in epidemiological analysis. Unfortunately, it is frequently missing in the PINNs results. Future developments of the proposed split PINN approach should consider UQ in more complex compartmental models, for example following the framework proposed by Linka et al. [40] which combines NNs and Bayesian inference.

This study focuses on the standard and simple SIR model. While the split approach can be easily adapted to more complex compartmental models (which, for example, consider an exposed compartment, deaths, re-infections, and vaccinations), the reduced equations described in (12) will require ad-hoc formulations depending on the model. It is important to underline that in this simple setting it would be possible to directly use purely data-driven DNNs to attempt predicting the future data as done in Case 7 (Fig 12). However a comparison between the results of PINN and DNN has not been presented in this manuscript because PINN has a wider goal with respect to DNN: PINNs in fact allow the user to also estimate and

predict the dynamics of model parameters and/or state variables that are not possible to link to the data if not using the model equations. This information is not available in more standard DNNs or statistical approaches such as the renewal equation, and it constitutes the main advantage of PINNs methods.

In conclusion, the proposed split PINN-based approach is a robust and easy-to-implement tool to monitor the initial spreading of a disease. It provides estimates of the temporal changes in the model parameters, which is essential to produce more accurate short-term forecasts.

## Supporting information

**S1 Text.** Appendix A. Neural Network architectures. Appendix B. Full SIR model with large data errors. Appendix C. Forecasting using the joint PINN approach. Appendix D. Comparison with the renewal equation.
(PDF)

## Acknowledgments

The authors are members of the Gruppo Nazionale Calcolo Scientifico—Istituto Nazionale di Alta Matematica (GNCS-INdAM) and C.M. is part of the project CUP_E53C23001670001. C. M. and M.F. acknowledge the support provided by BIRD2023 and ICEA Department of the University of Padova through the project "SurMoDeL: Deep Learning Surrogate Models for reservoir characterization". D.P. acknowledges the support provided by the PRIN project "Hydro-ROM, Reduced order models of hydraulic protection systems for extreme water hazards" H53D23001350006.

## Author Contributions

**Conceptualization:** Caterina Millevoi, Damiano Pasetto, Massimiliano Ferronato.

**Methodology:** Caterina Millevoi.

**Supervision:** Massimiliano Ferronato.

**Validation:** Damiano Pasetto.

**Visualization:** Caterina Millevoi.

**Writing – original draft:** Caterina Millevoi.

**Writing – review & editing:** Damiano Pasetto, Massimiliano Ferronato.

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
