## [Decision Letter · Decision Letter 0]

19 Feb 2024

Dear Dr Millevoi,

Thank you very much for submitting your manuscript "A Physics-Informed Neural Network approach for compartmental epidemiological models" for consideration at PLOS Computational Biology.

As with all papers reviewed by the journal, your manuscript was reviewed by members of the editorial board and by several independent reviewers. In light of the reviews (below this email), we would like to invite the resubmission of a significantly-revised version that takes into account the reviewers' comments.

I agree with both reviewers that this work potentially represents an important contribution to the field of computational epidemiology and infectious disease modeling; however, substantial work is needed to clarify why specific modeling techniques were selected and how the results may compare to more established methods. The authors should clarify whether true out-of-sample performance was measured (not necessarily prospectively, but at least on fully held-out data). If such out-of-sample testing was not performed doing so will be essential for fully addressing the reviewer comments. I also would like the authors to revise their work to make it more accessible to a broad audience. Please pay close attention to the detailed comments from the two reviewers.

We cannot make any decision about publication until we have seen the revised manuscript and your response to the reviewers' comments. Your revised manuscript is also likely to be sent to reviewers for further evaluation.

Sincerely,

Samuel V. Scarpino

Academic Editor

PLOS Computational Biology

Virginia Pitzer

Section Editor

PLOS Computational Biology

I agree with both reviewers that this work potentially represents an important contribution to the field of computational epidemiology and infectious disease modeling; however, substantial work is needed to clarify why specific modeling techniques were selected and how the results may compare to more established methods. The authors should clarify whether true out-of-sample performance was measured (not necessarily prospectively, but at least on fully held-out data). If such out-of-sample testing was not performed doing so will be essential for fully addressing the reviewer comments. I also would like the authors to revise their work to make it more accessible to a broad audience. Please pay close attention to the detailed comments from the two reviewers.

Reviewer's Responses to Questions

**Comments to the Authors:**

Reviewer #1: The paper introduces an innovative approach to address inverse problems related

to epidemiological data and utilizing PINNs. Specifically, two simple innovative

modifications, named the reduced and split approaches, are introduced

and compared in a comprehensive campaign of numerical results involving both

synthetic and realistic scenarios.

Strengths

• The authors have shown a clear understanding of the existing state-of-theart

about Machine Learning techniques for solving Inverse Problems and

in particular the application to the epidemiological context;

• The proposed methods introduce little modifications adapted to the epidemiological

framework to the well-established framework of PINNs.

Major Comments

• I believe there is a lack of explicit justification for employing Neural Networks

with large dimensions, as indicated in lines 369-373, particularly

regarding their depth and width. Could this choice be somehow linked to

the recent progress in infinitely-wide neural networks?

• It would be beneficial to include a thorough sensitivity analysis for the parameters

that define the topology of the neural network (like some comparison

between different architectures, even with smaller neurons or layer).

This is because the number of parameters used to calibrate PINNs’ weights

and biases significantly impacts the overall complexity of the training process.

• In lines 619–620, the authors assess that the proposed split PINN-based

approach is a robust and straightforward tool for monitoring the initial

spread of a disease. However, I am unable to evaluate the goodness of this

tool in the prediction sense, since, as I understand it, the numerical results

are retrospective. They approximate the dynamics within the same period

as the data used for training and have not been utilized to extrapolate

future behavior. Could you provide some clarification on this aspect?

Recommendation

I recommend that the paper addresses the previous points, and then it can be

accepted.

Reviewer #2: In this study, the authors apply a neural network approach informed by differential equations (Physics-Informed Neural Networks) to infer the parameters’ values that regulate the dynamics of an SIR model.

The results show that the approach adopted by the authors, based on a reduced-split of the training set, outperforms the methods that have been previously proposed in the literature both in terms of accuracy and computational times.

The study presents a few strengths and weaknesses at the same time, which I summarize in the following. Overall, the paper quality does not currently meet the standards to be published in PLOS Computational Biology and I think the paper requires a major revision before it can be considered for publication PCB.

I am suggesting a number of changes that I hope could improve the manuscript.

Strengths:

- The use of PINN in infectious disease modeling is promising – yet understudied – and the method proposed by the author improves on previous results, thus showing a more effective way of using PINN in this field.

- The study considers a range of application scenarios that include real-world data, and more realistic cases than previous studies.

Weaknesses:

- The main result of the study is the introduction of a novel training approach for PINNs applied to the SIR model. While I consider it interesting and useful, in terms of impact I am not sure it will be of high interest to the broad audience of PCB.

- For the readers who are not familiar with PINNs, it will be difficult to assess the quality of the results with respect to more traditional approaches. The paper is often technical, and it is hard to compare the results against more traditional infectious disease modeling techniques.

- The presentation of the results and the structure of the manuscript follows a bullet list style (Case 1, Case 2, etc.) that is not very engaging. Figures and tables repeat the same structure over and over, and they are sometimes hard to interpret.

- The paper suffers from an overly technical notation that can appear obscure to biologists, epidemiologists and infectious disease modellers.

Comments and suggestions

1. It would be important for infectious disease modelers to understand if PINNs provide benefits with respect to classic methods that are used to estimate Rt in real world cases. For instance, the widely adopted Rt estimation method proposed by Cori et al. AJE 2013. Can PINNs replace these approaches to some extent? In general, it is not obvious for those who work in IDM what tasks can be accomplished by PINN with better or similar performance than commonly used methods.

2. On a similar note, I did not fully grasp what is the portion of the epidemic curve that is used to train the PINN in each scenario. Does the model need the full curve as training input? In this case, its use would be strongly limited in the early phase of an outbreak.

3. Choosing a fixed value for the average infectious period (5 days) seems a strong assumption. Is this kept constant in all experiments? Often, there is a large uncertainty affecting estimates of the infectious period. How sensitive are the results to changes in the infectious period value?

4. The section Methods currently describes in detail the mathematical structure of the PINN. I would suggest extending the section with a description of the numerical methods that right now appear at the beginning of the results. All the information regarding the numerical experiments (size of training, testing, error metrics, etc.) should be part of the methods.

5. A graphic description of the numerical methods could also improve the readability of the paper. A summary figure describing how the epidemiological data are used in the training phase, what is the output of the model, etc.

6. I would recommend changing the structure of the manuscript with sections title that do not just report the names “Case 1, Case 2, Case 3…” For instance, Results could be grouped into two main parts. One considering synthetic data and the other considering COVID-19 data.

7. One major limitation of the PINN approach is to fit a deterministic SIR model to the data. The deterministic model leads to parameter estimates with almost exact values that are difficult to interpret when compared to traditional statistical methods used in the literature that provide a confidence interval of credible values. Is there a way to overcome this limitation by extending the range of possible outcomes of the model? I understand this is an open question that may lack a satisfying answer, but at least I would discuss this limitation more extensively at the end of the paper.

**Have the authors made all data and (if applicable) computational code underlying the findings in their manuscript fully available?**

Reviewer #1: Yes

Reviewer #2: Yes

PLOS authors have the option to publish the peer review history of their article (what does this mean?). If published, this will include your full peer review and any attached files.

Reviewer #1: No

Reviewer #2: No
---

## [Decision Letter · Decision Letter 1]

30 May 2024

Dear Dr Millevoi,

Thank you very much for submitting your manuscript "A Physics-Informed Neural Network approach for compartmental epidemiological models" for consideration at PLOS Computational Biology.

As with all papers reviewed by the journal, your manuscript was reviewed by members of the editorial board and by several independent reviewers. In light of the reviews (below this email), we would like to invite the resubmission of a significantly-revised version that takes into account the reviewers' comments.

I appreciate the authors' effort on the revisions and agree with the reviewers that the manuscript has been strengthened as a result. I agree with Reviewer 3's comments that the work needs some improvements. In particular, the authors should pay careful attention to addressing concerns about whether regularization is sufficient, whether it is appropriate to use a Python package that is core to the method but is not longer being actively maintained, whether off-the-shelf deep neural networks could perform as well or better than what is proposed here, and whether the forecasting objectives are sufficiently complicated to meaningfully test the approach. I also agree that there are many places where the methodology can be more clearly specified. Lastly, the Reviewers raised concerns about whether all necessary code/data is available to reproduce the entire study. If the authors cannot provide all necessary code/data they must formally request a waiver of the Journal's policies, which I cannot promise will be granted. I believe this work is interesting and could be impactful, so I hope that authors pay careful attention to the detailed comments from Reviewer 3 and address the code/data availability concerns.

We cannot make any decision about publication until we have seen the revised manuscript and your response to the reviewers' comments. Your revised manuscript is also likely to be sent to reviewers for further evaluation.

Sincerely,

Samuel V. Scarpino

Academic Editor

PLOS Computational Biology

Virginia Pitzer

Section Editor

PLOS Computational Biology

I appreciate the authors' effort on the revisions and agree with the reviewers that the manuscript has been strengthened as a result. I agree with Reviewer 3's comments that the work needs some improvements. In particular, the authors should pay careful attention to addressing concerns about whether regularization is sufficient, whether it is appropriate to use a Python package that is core to the method but is not longer being actively maintained, whether off-the-shelf deep neural networks could perform as well or better than what is proposed here, and whether the forecasting objectives are sufficiently complicated to meaningfully test the approach. I also agree that there are many places where the methodology can be more clearly specified. Lastly, the Reviewers raised concerns about whether all necessary code/data is available to reproduce the entire study. If the authors cannot provide all necessary code/data they must formally request a waive of the Journal's policies, which I cannot promise will be granted. I believe this work is interesting and could be impactful, so I hope that authors pay careful attention to the detailed comments from Reviewer 3 and address the code/data availability concerns.

Reviewer's Responses to Questions

**Comments to the Authors:**

Reviewer #1: This work has strongly improved after the revision and it represents a significant contribution about the interplay of Machine Learning Schemes and epidemiology.

By my side, all the raised questions have been answered.

Reviewer #2: I thank the authors for revising the manuscript in light of my remarks. All my comments have been addressed.

Reviewer #3: 

The authors propose the use of PINNs to regress time-dependent epidemiological parameters from data. They propose two training strategies called joint and split and also propose a method independent of the initial condition where the initial conditions do not enter the loss function. The article has been thoroughly reviewed before and the epidemiological models used are time-tested and well established in the literature. The innovation proposed by the authors is in the design and training of the PINNs. To do this the authors use a publicly available Python module called SciANN based on Tensorflow. After the first round of reviews, the authors added an example of forecasting on held-out data. And improved several parts of the work and the draft. However, I still find the draft lacking in some respects, and the methods used by the authors are not sufficient.

While the core idea proposed by the authors is interesting, the implementation needs more rigor and some of the conclusions drawn by the authors might be affected by this lack of rigor. The authors introduce and use concepts that are not made clear to the audience this is aimed at and the training of the NN models has not been done following best-practices from proper ablation studies. The authors have added some studies of the NN architecture in the supplementary document but thia is not sufficient. The authors have not explored learning rate schedules, regularization, or dependence on activation functions.

Comments:

- In lines 423-425, the authors mention calibrating the weights in the loss function using eigenvalues of the NTK and cite a reference. The authors do not put any details on what they do to calibrate the weights using an NTK. From the code, it seems like they use a built-in function in the SciANN module to do this. However, they do not show whether their errors improve by using the NTK calibration or not. I am guessing it might help with the solution of the inverse problem but not with the out-of-sample prediction.

- From the supplementary material on the NN ablation study, it is not clear why the authors get larger errors for models of higher complexity. The authors seem to train the models on a fixed number of epochs. If they do so, more complex models are bound to overfit as the authors do not mention using any explicit regularization of the weights of the NN. It is straightforward to understand that NNs of greater complexity will be prone to overfitting if no implicit or explicit regularization is used. I would suggest the authors use early stopping as an implicit regularizer and/or add regularization to the loss function and check how that affects the NN training. This can potentially reduce training times too.

- The actual time taken by the NNs to train is not meaningful as the authors do not specify the computer architecture they use. The code provided by the authors is not well documented and is a single Python notebook that does not allow one to reproduce all the results. It looks more like a sample. Moreover, the authors use the package SciANN which is limited to Python <= v3.10 and works only with Tensorflow < v2.11. The training times and accuracy of the NN can be highly dependent on the Tensorflow version. The authors do not specify which version of SciANN/Tensorflow they use. If the authors want their work to be useful in the future they should write a code that can be used without much trouble in the future. The authors could try implementing a PINN in Tensorflow or Pytorch. I know the objective of the paper is about training strategies, but many of the conclusions the authors come to can be dependent on the fact that they use a deprecated framework for their NNs (SciANN is no longer maintained as of 02/2023).

- In SI S3, the authors show that using the renewal equation approach of Cori et al. leads to comparable or better results than using PINNs. Given the complexity of the approach proposed by the authors, what is the real advantage of using their method other than just applying machine learning to this problem?

- In their response, the authors emphasize: “A final important consideration is that PINNs benefit from the so-called ‘transfer learning’ typical of Neural Networks, that is the possibility to adapt an already trained NN to new data. This has been explored, for example, in the application to forecasting (Section 2.2.2).” The authors are not clear about what they call “transfer learning”. It's a very loaded term and can mean several different things. The authors do not specify their strategy for transfer learning. Do they freeze the weights and add layers to the NN or do they continue the training with the same architecture as before just adding more data? If it is the latter, then it is not really transfer learning as data from the same domain and modality is being used for further training the same architecture without any fixed weights. I suspect that without using their version of “transfer learning” the NNs would do equally well leading to a lack of reason for using this transfer learning method. Please refer to A Comprehensive Survey on Transfer Learning by Zhuang et al. for more details. I would strongly recommend the authors to not incorrectly refer to their PINNS using transfer learning.

- The authors claim: “As one could expect, the projections are not accurate when few data are available or when the model is close to the peak of the epidemic (Figs. 12a and 12b). Good outcomes are achieved during the recession (Figs. 12c and 12f).” How is the approach proposed in this paper better than that proposed in Emergence of universality in the transmission dynamics of COVID-19 by Paul et al. or any other models that perform significantly better during recession than during the growth of the epidemic? If not, then the PINN approach proposed in this work is, at best, a descriptive approach and not a predictive approach which makes the use of PINNs in this case a bit questionable.

- I do not understand why the authors create their own accuracy metric (eq. 36) rather than using standardized ones like R^2 which has a very clear interpretation.

- The simulated scenarios are very simple and I suspect even a DNN can be used to find the solution. The authors do not demonstrate that their PINN outperforms a simple DNN implementation.

- I do not see a comparison between split and reduced-split nor where the reduced-split method significantly outperforms the split method. The code does not seem to have any examples of the two separately.

- For the case of split training, it seems like the SciANN module is using a reduce-on-plateau learning rate schedule where the learning rate is reduced by a factor of 2 on plateau of the validation loss. Hence, the authors’ statement in section 1.6 that they use a learning rate of 0.001 in all cases is incorrect. Secondly, it is well known that learning rate schedules significantly improve training. I’d encourage the authors to explore the use of learning rate schedules in training the PINNs.

- Using Tensorflow 2.11 and Keras 2.11 I get an order of magnitude smaller errors that the authors get with the code they have shared in their repository. Since the authors did not specify the versions of the modules they used, it is impossible for me to say whether the improvement is due to the module version or the initialization of random numbers (unlikely) that have not been seeded in the provided notebook.

- The authors have provided no insight as to why the split method works better than the join. In a real-world scenario, it might be imaginable that the infection rate is affected by real-world uncertainties like testing strategies etc., in which case a data-driven approach to this estimating the infection rate might work better. However, for simulated data, this is not the case. So, some insights into why split training works better would be useful.

The fork of the repository run with TensorFlow 2.11.0 can be found here where the discrepancy in error estimates can be verified (the requirements file specifies the module versions): https://github.com/anonymousacademicc/EpiPINN

**Have the authors made all data and (if applicable) computational code underlying the findings in their manuscript fully available?**

Reviewer #1: None

Reviewer #2: Yes

Reviewer #3: **No: **The authors have only provided sample code and not all the code necessary to reproduce their work. They have specified a link to the data but have not provided code for the ETL of the data. The link they provide simply leads to a landing page: https://www.epicentro.iss.it/en/coronavirus/. With the information and the code that they provide it is impossible to reproduce their results more so because they have not specified the version of the python modules used. Moreover, they use a python module that is no longer supported by the developers and is deprecated. The module depends on interpreters and modules that will be deprecated in the near future limiting the utility of the computational method that they develop and inhibiting the reproducibility of their work.

PLOS authors have the option to publish the peer review history of their article (what does this mean?). If published, this will include your full peer review and any attached files.

Reviewer #1: No

Reviewer #2: No

Reviewer #3: No
---

## [Decision Letter · Decision Letter 2]

5 Aug 2024

Dear Dr Millevoi,

We are pleased to inform you that your manuscript 'A Physics-Informed Neural Network approach for compartmental epidemiological models' has been provisionally accepted for publication in PLOS Computational Biology.

The reviewer's comments are included below for your consideration, although we feel they are beyond the scope of the current analysis and do not need to be addressed at this time.

Best regards,

Samuel V. Scarpino

Academic Editor

PLOS Computational Biology

Virginia Pitzer

Section Editor

PLOS Computational Biology

Reviewer's Responses to Questions

**Comments to the Authors:**

Reviewer #3: I thank the authors for the detailed response to the review and the corrections. I have a few major concerns

1. The authors say that there are three sources of errors in an NN. I agree. The last two that they cite in that list can be mitigated by regularization. Yet, the authors claim that regularization is unnecessary for PINNs. I am not sure the two statements are not contradictory. Yes, it is true that certain implementations of PINN show a reduced necessity for regularizers. However this is not true for any and every PINN regardless of their complexity. While the additional terms in the loss function can help regularize the neural network, even in the paper the authors cite, it does not say that regularization is not necessary. Unfortunately, in the world of neural networks such things can only be confirmed by empirical evidence. The authors should show that it is not necessary if they wish to claim so.

2. I disagree with the authors that a computational paper proposing a new neural network based method can be published without supporting code to reproduce their work. I understand that dependencies tend to get deprecated over time. However, it is not acceptable that reproducing the work is tedious even during the review process a few months after the completion of the work. I maintain that the burden of reproducibility of the work and the creating of a code base that others can further develop lies on the authors. The early retirement of a lead developer of a code not maintained by the authors cannot be an excuse for the reproduction of the work being unnecessarily painful.

3. The authors claim that forecasting is not their goal. I can accept that. Then I do not understand why it is presented in the paper even though I do understand that the previous reviewers asked for it. It can be simply stated as being something outside the scope of the work. If the goal is just parameter estimation, I do not see how this approach is doing much better than other approaches. The authors themselves show that the renewal equation works as well. The authors make four cases for the advantage of PINNs over the renewal model but none of them can lead to significantly better or quicker estimation of parameters. Also, the authors already admitted that forecasting is not the objective. I do not see what advantage their model is providing. Even if the authors could show that they are doing better than a simple DNN, it might have held some credibility. I understand that a simple DNN would miss out on the underlying epidemiological dynamics, but then the entire point of using a simple DNN is to see how well one can do without using physics information, i.e., how much better a DNN can do when knowledge of the dynamics is added.

Given these reasons, I do not find a compelling case for recommending this paper for publication.

**Have the authors made all data and (if applicable) computational code underlying the findings in their manuscript fully available?**

Reviewer #3: Yes

PLOS authors have the option to publish the peer review history of their article (what does this mean?). If published, this will include your full peer review and any attached files.

Reviewer #3: **Yes: **Ayan Paul

---

## [Editor Report · Acceptance letter]

28 Aug 2024

PCOMPBIOL-D-23-02037R2 

A Physics-Informed Neural Network approach for compartmental epidemiological models

Dear Dr Millevoi,

I am pleased to inform you that your manuscript has been formally accepted for publication in PLOS Computational Biology. Your manuscript is now with our production department and you will be notified of the publication date in due course.

With kind regards,

Anita Estes
